# EGPlace: An Efficient Macro Placement Method via Evolutionary Search with Greedy Repositioning Guided Mutation

Ji Deng [1]   Zhao Li [2]   Ji Zhang [3]   Jun Gao [1]

## Abstract

Macro placement, which involves optimizing the positions of modules, is a critical phase in modern integrated circuit design and significantly influences chip performance. The growing complexity of integrated circuits demands increasingly sophisticated placement solutions. Existing approaches have evolved along two primary paths (e.g., constructive and adjustment methods), but they face significant practical limitations that affect real-world chip design. Recent hybrid frameworks such as WireMask-EA have attempted to combine these strategies, but significant technical barriers still remain, including the computational overhead from separated layout adjustment and reconstruction that often require complete layout rebuilding, the inefficient exploration of design spaces due to random mutation operations, and the computational complexity of mask-based construction methods that limit scalability. To overcome these limitations, we introduce EGPlace, a novel evolutionary optimization framework that combines guided mutation strategies with efficient layout reconstruction. EGPlace introduces two key innovations: a greedy repositioning-guided mutation operator that systematically identifies and optimizes critical layout regions, and an efficient mask computation algorithm that accelerates layout evaluation. Our extensive evaluation using ISPD2005 and Ariane RISC-V CPU benchmarks demonstrate that EGPlace reduces wirelength by **10.8%** and **9.3%** compared to WireMask-EA and the state-of-the-art reinforcement learning-based constructive method EfficientPlace, respectively, while achieving speedups of **7.8×** and **2.8×** over these methods.

[1]School of Computer Science, Peking University, BeiJing, China [2]Zhejiang Lab, Zhejiang, China [3]Nanjing University of Aeronautics and Astronautics, China. Correspondence to: Jun Gao <gaojun@pku.edu.cn>.

*Proceedings of the 42$^{st}$ International Conference on Machine Learning*, Vancouver, Canada. PMLR 267, 2025. Copyright 2025 by the author(s).

## 1. Introduction

The exponential growth in integrated circuit complexity has made chip placement a critical challenge in electronic design automation (MacMillen et al., 2000). The task involves positioning thousands of macro blocks and millions of standard cells on a chip, creating a solution space even more complex than advanced strategic games like Go (Mirhoseini et al., 2021). Research has been focused on macro placement optimization (Geng et al., 2024b; Mirhoseini et al., 2021; Lai et al., 2022; Shi et al., 2023) as these functional blocks significantly impact overall circuit performance.

The macro placement problem presents a complex optimization problem with three interrelated technical objectives that determine circuit viability (Chen et al., 2006). Minimizing total wirelength between connected models reduces signal delays and improves chip performance, while managing routing congestion ensures adequate resources for signal transmission (Wang et al., 2009). The complete elimination of physical overlaps between models remains essential for manufacturing feasibility (Lai et al., 2022). The technical complexity emerges from the nonlinear relationships between model positions and performance metrics, where local modifications can have significant effects on global optimization goals. This challenge compounds with the intricate connections between models, which together create an enormous combinatorial solution space. Achieving high-quality placement solutions requires simultaneously balancing these competing objectives and addressing the complex challenges.

Research in macro placement optimization has developed two different categories of approaches: *adjustment-based methods* that iteratively improve existing layouts, and *constructive methods* that build solutions progressively. The adjustment-based category includes both stochastic and analytical optimization techniques. Stochastic methods such as simulated annealing (Sechen & Sangiovanni-Vincentelli, 1985) and evolutionary algorithms (Cohoon & Paris, 1987; Shahookar & Mazumder, 1991; Kling & Banerjee, 1989) modify layouts through random perturbations to explore diverse solutions. However, this undirected exploration often leads to low sample efficiency. Analytical approaches (Lu et al., 2015; Cheng et al., 2018; Kahng et al., 2005) trans-

form the placement problem into mathematical equations solved through techniques like gradient descent. These methods face practical limitations from relaxing overlap constraints and difficulty in handling non-differentiable congestion metrics. Constructive methods build layouts from scratch using different strategies. Recent advances on constructive methods have focused on reinforcement learning approaches (Mirhoseini et al., 2021; Cheng et al., 2022; Lai et al., 2022; Geng et al., 2024b) that model placement as a Markov Decision Process with sequential model positioning. While these methods have shown promising results, they require substantial computational resources for training. The sequential nature of their decision-making also means each placement choice lacks sufficient reception field about future layout configurations.

The integration of constructive and adjustment methods leads to promising approaches to macro placement optimization. A recent state-of-the-art method WireMask-EA (Shi et al., 2023) implements this concept through an evolutionary algorithm framework. The method applies random mutations to layouts and uses greedy reconstruction to improve solution quality. However, the hybrid approaches still encounter three major technical challenges: Executing layout adjustments and reconstructions sequentially can lead to inefficiencies, as local changes may require a complete layout overhaul; random mutations in stochastic optimization can waste computational resources by exploring less promising configurations; the mask computations needed for assessing module positions can become a performance bottleneck due to redundant calculations during placement.

To solve the aforementioned technical challenges, we introduce EGPlace, an evolutionary optimization framework for macro placement that integrates guided mutation with efficient reconstruction techniques. Unlike previous approaches that separate adjustment and construction phases (Shi et al., 2023), EGPlace implements a unified mutation operator that selects modules for repositioning based on their quality impact scores and employs an efficient greedy strategy for module relocation. To enhance sample efficiency during exploration within the vast search space, the framework incorporates guidance mechanisms at multiple levels - using fitness functions to identify promising layouts for evolution and prioritizing poorly placed modules for repositioning. This targeted optimization approach, combined with an optimized mask computation algorithm, enables EGPlace to achieve superior results with significantly reduced computational requirements. The main contributions of this work include:

- A novel and efficient mutation operator with a guidance mechanism that identifies critical modules for repositioning based on their impact on overall layout quality, thereby improving sample efficiency. More-

over, this mutation operator seamlessly integrates layout adjustment and reconstruction phases, eliminating the computational overhead caused by full layout reconstruction after local modifications.

- An advanced mask computation algorithm that reduces the computational complexity from quadratic to linear time per module. This improvement enables rapid evaluation of potential module positions and accelerates the overall optimization process.

- Extensive experimental validation on standard ISPD2005 benchmarks demonstrates that EGPlace achieves 11% better Half Perimeter Wire Length (HPWL) (Chen et al., 2006) than WireMask-EA (Shi et al., 2023), 9% improvement over EfficientPlace (Geng et al., 2024b), and a 7.8× speedup compared to WireMask-EA.

## 2. Related Work

Current chip placement methods are maninly classified into constructive and adjustment-based techniques. In a significant development, recent research has focused on combining these approaches to create more effective hybrid solutions.

**Constructive Methods.** Constructive methods generate layouts from scratch. Early constructive methods rely on heuristic rules, such as greedy placement rule (Magnuson, 1977; Fukunaga et al., 1983). Those methods place interconnected modules in proximity to reduce wirelength. Although they are efficient, it is challenging to determine rules that perform well across different chip benchmarks. Breuer et al. introduces partition-based methods, which divide circuits into clusters to minimize inter-cluster connections and place them in corresponding subregions (Breuer, 1977). Later improvements include terminal propagation to account for inter-cluster connections (Dunlop & Kernighan, 1985) and advanced hypergraph partitioning techniques (Fiduccia & Mattheyses, 1988; Karypis et al., 1997). However, these methods optimize placements within each partition independently, limiting overall performance.

Recent work utilizes Reinforcement Leaning (RL) to learn constructive placement strategies. Mirhoseini et al. (Mirhoseini et al., 2021) introduced sequential module placement using a trained RL agent. Many studies have focused on improving the performance of RL-based methods. PRNet (Cheng & Yan, 2021) integrated chip routing considerations to refine reward signals. MaskPlace (Lai et al., 2022) leveraged CNNs to better capture pin location information. Subsequent work prioritizes computational efficiency. Chipformer (Lai et al., 2023) employed pretrained decision transformers (Zheng et al., 2022) for rapid adaptation to new circuits. EfficientPlace (Geng et al., 2024b) further advanced efficiency through a learning-inside-optimization

framework that combines state storage with Monte Carlo Tree Search guidance. Despite these innovations, existing RL-based constructive approaches face two persistent limitations: the substantial computational overhead of agent training and the inherent myopia of sequential placement decisions, where each choice does not sufficiently consider future module positions.

**Adjustment-based Methods.** Adjustment-based methods iteratively adjust module locations to improve layout quality. These methods include stochastic-based and analytical-based approaches. Stochastic-based approaches typically use simulated annealing algorithms and represent layouts with data structures like sequence pairs (Murata et al., 2002), B*-trees (Chang et al., 2000) and corner block lists (Hong et al., 2000). These approaches apply perturbations to their data structures to explore the search space, then transform them back to the original layout representation for evaluation. However, such transformations are computationally expensive (Shi et al., 2023), while random perturbations often lead to low sample efficiency. Analytical-based methods, including quadratic (Viswanathan & Chu, 2005) and nonlinear approaches (Cheng et al., 2018; Lin et al., 2019; Lu et al., 2015), model optimization objectives with analytical equations. Nonlinear methods like ePlace (Lu et al., 2015) treat module density as charge density in an electrostatic system and have shown strong performance. Analytical-based methods often treat overlap as a soft constraint, addressing it during a subsequent legalization phase, which may lead to a significant increase in wirelength or even fail to generate valid layouts. They also ignore congestion, which is nondifferentiable and hard to optimize with gradient descent. Some recent RL-based methods (Xue et al., 2024; Chiang et al., 2025) adopt a novel adjustment paradigm, where placement begins with a complete layout and policies are learned to adjust the position of one or a group of modules at each step. While this approach benefits from a comprehensive state representation that captures the full layout context, it incurs considerable computational overhead.

**Hybrid Methods.** Some recent research has focused on synthesizing the strengths of constructive and adjustment methods in macro placement optimization. For instance, WireMask-EA (Shi et al., 2023) employed a simple (1+1)-EA (Zhou et al., 2019) as its backbone, which maintains a population of only one layout. In each iteration WireMask-EA applies random module swapping adjustments on the layout to explore the search space, and reposition modules with a greedy constructive method to improve the quality. It takes advantage of both the efficiency of greedy placement methods and the capability to broadly explore the solution space of adjustment methods, and achieves better layout quality than many state-of-the-art RL-based methods. However, despite its innovative design, the method still confronts several computational and optimization challenges that were

outlined in the introduction. In addition, the recent work LaMPlace (Geng et al., 2024a) integrates a learned mask, designed to reflect final Power, Performance and Area (PPA) objectives, into the WireMask-EA framework, and use it in place of the original WireMask to guide macro placement. This enhancement leads to substantial improvements in PPA performance. The learned mask from LaMPlace can also be incorporated into EGPlace to potientially improve the final layout quality.

## 3. Problem Description & Overall Framework

The input of macro placement comprises a rectangular chip substrate and a circuit netlist represented as a hypergraph, where modules serve as nodes while nets function as hyperedges that connect the pins located on different modules.

Layout quality evaluation involves three key metrics. The primary metric, wirelength, measures total routing resources and affects circuit performance. Since the exact wirelength can be determined only after routing, a more efficient approximation, Half Perimeter Wire Length (HPWL) (Chen et al., 2006), is typically used, calculated from the sum of half-perimeters of net-bounding boxes. For routing congestion analysis, the Rectangular Uniform wire DensitY (RUDY) metric (Spindler & Johannes, 2007) gauges resource utilization density. Additionally, minimizing physical overlap between modules, quantified by an overlap rate (Lai et al., 2022), is crucial for manufacturability. While ideal layouts aim for zero overlap, practical methods (Lin et al., 2019; Lai et al., 2022) may relax this constraint to improve optimization efficiency, especially on benchmarks where modules occupy a large proportion of the placement region, such as the "ariane" benchmark in Appendix D.4. Detailed metric definitions and calculations are found in Appendix B.

The optimization objective combines these metrics through a weighted sum formulation shown in Equation (1). The hyperparameters $\lambda_1$ and $\lambda_2$ control the relative importance of RUDY and overlap penalties compared to the primary HPWL objective, to produce the layouts that balance competing design constraints:

$$\text{Objective} = \text{HPWL} + \lambda_1 \cdot \text{RUDY} + \lambda_2 \cdot \text{overlap} \quad (1)$$

In this paper, we propose EGPlace for efficient macro placement. The framework of EGPlace is illustrated in Fig. 1. It utilizes an evolutionary backbone combined with greedy repositioning-guided mutation.

## 4. Evolutionary Backbone

EGPlace employs evolutionary search as its optimization backbone due to its effectiveness in handling complex, nondifferentiable objectives and discrete optimization problems.

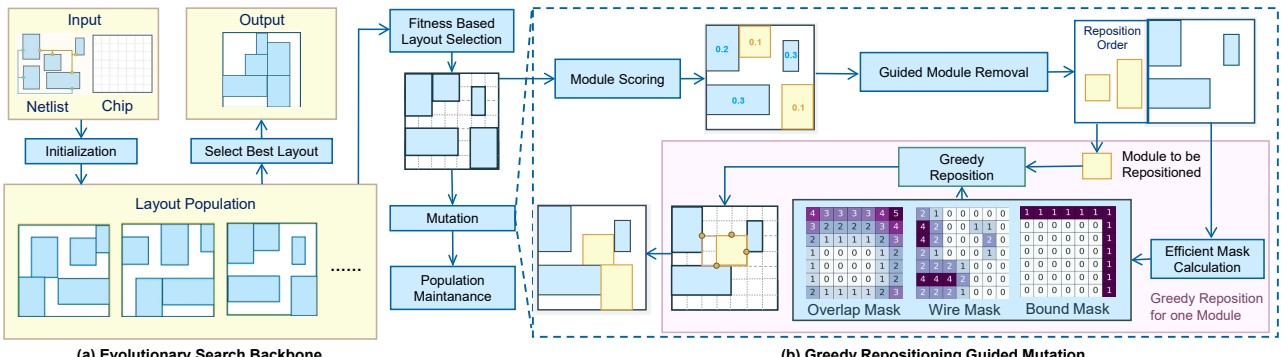

(a) Evolutionary Search Backbone      (b) Greedy Repositioning Guided Mutation

*Figure 1.* **EGPlace Framework.** We employ an evolutionary search backbone and propose a greedy repositioning-guided mutation. The mutation process operates in two phases. In the first phase, modules are selected for repositioning based on scores that measure their impact on layout quality. In the second phase, a mask-guided greedy strategy is applied to determine new positions for selected modules.

The population-based approach can also naturally maintain solution diversity while exploring the vast configuration space of circuit layouts.

The search evolves a fixed-size population of layouts through successive iterations. By maintaining a bounded population, we avoid memory and runtime overhead while preserving solution diversity, critical for escaping local optima in macro placement. During each iteration, a layout undergoes the proposed mutation operator, which integrates both constructive and adjustment strategies. We focus exclusively on mutation operations as our experiments showed that crossover mechanisms often disrupt critical local optimization patterns in circuit placement. The modified layout may enter the population based on its fitness evaluation, replacing the lowest-performing existing solution. This selective pressure drives the population toward optimized configurations until reaching the specified iteration limit.

**Initialization.** The population is initialized with layouts constructed using the greedy placement strategy described in Section 5.2. This constructive strategy is highly efficient and generates promising layouts for evolution.

**Fitness-based Layout Selection.** The fitness function evaluates layout quality and guides layout selection. The fitness $f_{L_i}$ of layout $L_i$ is defined as the negative of the optimization objective in Equation (1), where higher fitness values correspond to better layout quality. The weights $\lambda_1$ and $\lambda_2$ can be adjusted by the user to obtain layouts with congestion and overlap below desired thresholds.

A layout $L_i$ is selected from the population with a probability $p(L_i)$ positively correlated to its fitness, as defined in Equation (2), where $P$ denotes the layout population.

$$p(L_i) = \frac{e^{f_{L_i}}}{\sum_{L_j \in P} e^{f_{L_j}}} \qquad (2)$$

This fitness-guided selection is based on the intuition that layouts with higher fitness are more likely to result in high-quality layouts after adjustment. It ensures a more effective search while still allowing for random exploration.

**Layout Maintenance.** The selected layout undergoes mutation as described in Section 5 to generate a new layout. Then, the mutated layout is added to the population. If the population size exceeds the specified limit, the layout with the lowest fitness is discarded. This helps maintain a manageable population size and improves efficiency by stopping the mutation of less promising layouts.

## 5. Greedy Repositioning Guided Mutation

Our proposed mutation operator builds on three key design principles: unified integration of construction and adjustment phases, guided mutation processes that maintain beneficial randomness, and computationally efficient implementation of masks for rapid iteration.

Guided by the above principles, mutation operator functions through a two-phase process. In the selection phase, modules receive quality scores based on their impact on overall layout metrics. Modules with poorer placement quality have higher selection probability for repositioning, focusing computational resources on the most promising improvement opportunities. The repositioning phase then applies a greedy strategy to the selected modules, determining new positions that optimize local quality metrics while maintaining global layout coherence. The following sections detail these selection and repositioning mechanisms that form the core of our mutation strategy.

---

**Algorithm 1** Greedy Placement Strategy

---

**Input:** The placement area divided into $B \times B$ bins, the netlist $N$, and the set of modules $M$ selected to be placed. During population initialization, $M$ includes all macros. In greedy repositioning, $M$ consists of macros selected according to Section 5.1.
**Output:** Layout after greedy placement

1: **if** $|M|$ = the total number of modules **then**
2:     Sort the modules according to their size and connection, and store the results in the list $L$
3: **end if**
4: **while** $|L| > 0$ **do**
5:     $m \leftarrow$ the first module from $L$
6:     Compute $WireMask$, $OverlapMask$ and $BoundMask$ for $m$
7:     Select the set of valid bins $B_{val}$,
    s.t. for each bin $b \in B_{val}$, $BoundMask[b] = 1$ and $OverlapMask[b] = min(OverlapMask)$
8:     Select a set of bins for greedy placement $B_{greedy}$ from $B_{val}$,
    where $\forall b \in B_{greedy}, b \in B_{val} \land WireMask[b_{greedy}] = min(WireMask)$
9:     Randomly select a bin $b_{greedy} \in B_{greedy}$
10:    Place the bottom left corner of $m$ into $b_{greedy}$
11:    Remove $m$ from $L$
12: **end while**

**Return:** A layout

---

## 5.1. Module Selection for Repositioning

We perform guided module selection to enhance performance, where poorly placed modules are more likely to be removed and repositioned. This is achieved by scoring modules according to their impact on layout quality, with the removal probabilities proportional to their scores.

For a given module $m$, its scoring function is defined as a weighted sum of three terms that reflect its influence on HPWL, congestion, and overlap, respectively, as shown in Equation (3). Each term undergoes normalization before being combined to compute the module score.

$$score_m = wirelen_m + \lambda_1 \cdot cong_m + \lambda_2 \cdot \frac{overlap_m}{l_x^m \cdot l_y^m} \quad (3)$$

The first term, $wirelen_m$, represents the module $m$'s impact on the wirelength. It is defined as the average Manhattan distance from each pin $p$ of the module to the center of the bounding box of the net that contains the pin, as shown in Equation (4), where $P_m$ denotes the set of pins on module $m$, and $e_p$ denotes the center of the bounding box of the net that contains pin $p$. A larger $wirelen_m$ implies that the module is placed farther from the center, increasing the likelihood of a longer wirelength.

$$wirelen_m = \frac{1}{|P_m|} \sum_{p \in P_m} d(p, e_p) \quad (4)$$

The second term, $cong_m$, measures the module's impact on congestion. We use RUDY as the congestion metric. The RUDY value of the layout is defined as the maximum RUDY value among all bins in the placement region (Spindler & Johannes, 2007). Since RUDY is determined by the widths

and heights of net bounding boxes, we quantify module $m$'s contribution to congestion by counting the number of nets that contain pins on $m$ and significantly contribute to layout congestion. Specifically, the RUDY value of a net is defined as the maximum RUDY value of the bins covered by its bounding box. We compute $cong_m$ as the number of pins on $m$ belonging to nets whose RUDY values exceed $r\%$ of the overall layout's RUDY value, where $r\%$ is a hyperparameter set to a high value (e.g. 98%). The formal definition is given in Equation (5), where $rudy(E_p)$ and $rudy(L)$ represent the RUDY value of the net containing pin $p$ and the RUDY value of the entire layout $L$, respectively.

$$cong_m = \sum_{p \in P_m} \mathbf{1}_{rudy(E_p) > r\% \cdot rudy(L)} \quad (5)$$

The third term quantifies the overlapped area of $m$ with other modules, normalized by the size of $m$.

Subsequently, a specified ratio of modules is selected for removal and repositioning, with the selection probability being proportional to their respective scores. The selection probability of module $m$ is calculated by Equation 6, where $M$ denotes the set of all modules.

$$p_m^{select} = \frac{score_m}{\sum_{i \in M} score_i} \quad (6)$$

## 5.2. Greedy Reposition with Efficient Mask Calculation

The greedy placement strategy primarily optimizes the HPWL and overlap metric, while the congestion metric is mitigated by identifying and repositioning modules that significantly contribute to congestion. The greedy placement strategy places modules in valid locations that minimize overlap while keeping them close to their connected counterparts to optimize HPWL. We employ this strategy for module repositioning because it is highly efficient and has

been shown to yield promising results in the previous work (Shi et al., 2023).

The sequence of module placement substantially affects the final layout quality (Lai et al., 2022; Shi et al., 2023; Geng et al., 2024b). Our method orders module placement by both size and connectivity, placing larger and highly connected modules first. This strategy proves effective because larger modules constrain future placement options, and highly connected modules influence many other components through their wire dependencies.

To determine the placement location for a module, we compute masks to evaluate the quality of its placement at each position. Following previous methods (Lai et al., 2022; Shi et al., 2023; Geng et al., 2024b), we use wire masks to evaluate the increase in HPWL and bound masks to identify whether the module exceeds the chip boundaries. We further introduce overlap masks to quantify the additional overlap introduced by placing the module. Overlap masks calculate the exact increase in overlap between models, which differs from previous methods that use binary values to indicate whether placing a module at each position causes overlap. This helps reduce overlaps in scenarios with large module area coverage, where placement methods struggle to completely eliminate module overlap, such as in the "ariane" benchmark in Appendix D.4.

Guided by these masks, the greedy strategy selects the valid position for each module that minimizes both HPWL and overlap increase. The valid positions are those that do not cause boundary violations. A detailed overview of this process is provided in Algorithm 1. Note that the population is also initialized using greedy placement, where the set of modules to be placed includes all modules in the circuit. When considering the placement of module $m$ in Fig. 1, the meaning of the masks and how the placement location is chosen based on these masks are shown in Fig. 2.

**Efficient Mask Computation.** A straightforward implementation of mask computation requires iterating over all bins in the placement area, resulting in a time complexity proportional to $B^2$, where $B$ is the number of bins per row in the placement region. Lai et al. (Lai et al., 2022) has provided an efficient method for computing wire masks and bound masks.

We further propose an efficient algorithm for computing overlap masks to facilitate fast module repositioning. We compute the overlap between the current module $m$ and each placed module to derive the overlap mask before placing module $m$. The efficient computation of overlap between two modules $m$ and $n$ is demonstrated in Fig. 2. We exploit the linearity of overlap between modules $m$ and $n$ along both the $x$- and $y$-axes to compute overlap in linear time. As module $m$ moves across the placement region, the overlap

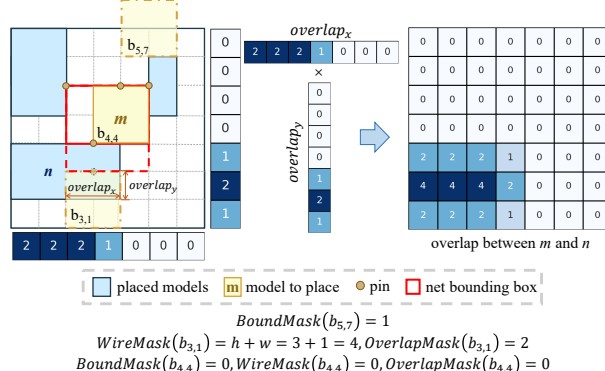

*Figure 2.* **Demonstration of Mask Calculation in Module Placement.** When module $m$ is placed in bin $b_{5,7}$, it exceeds the placement region, so the bound mask is set to 1. When $m$ is placed in $b_{3,1}$, the wire mask is determined by the increase in the half-perimeter of the net bounding box, while the overlap mask is computed by summing the overlapping area between $m$ and previously placed modules. The overlap between two modules $m$ and $n$ is efficiently calculated in linear time along the $x$- and $y$-directions and combined using an outer product. Placing $m$ in $b_{4,4}$ results in a bound mask of 0, no increase in overlap, and minimal HPWL growth. Therefore, $b_{4,4}$ is selected by the greedy strategy.

in one direction increases linearly, stays constant over a range, and then decreases linearly. This allows the overlap along the $x$- direction to be computed in linear time. The same approach can be applied to calculate the overlap in the $y$-direction. The overlap results from both directions are combined using the outer product to compute the total overlap between the two modules. The overall overlap mask is obtained by summing the overlaps between module $m$ and all placed modules. Suppose $m$ is the $i$-th module to be placed, this computation has a time complexity of $O(iB)$, which improves upon the $O(iB^2)$ complexity of the straightforward implementation. A detailed computation of module overlap is provided in Algorithm 2 in the Appendix.

## 6. Experiments

**Benchmarks, Baselines and Settings.** Our experimental evaluation uses eight benchmark circuits from the widely used ISPD2005 dataset (Nam et al., 2005) and one circuit from the Ariane RISC-V CPU design benchmark (Zaruba & Benini, 2019). The ISPD2005 benchmarks have been extensively used in previous macro placement research (Lai et al., 2022; 2023; Geng et al., 2024b), which enables direct performance comparisons with existing methods. The Ariane benchmark provides an important real-world test case. Dataset characteristics are described in Appendix D.1. We benchmark EGPlace against leading placement methods, including reinforcement learning approaches GraphPlace (Mirhoseini et al., 2021), DeepPR

*Table 1.* **HPWL Values ($\times 10^5$) Achieved by Different Macro Placement Methods on the ISPD2005 Dataset** The results of baseline methods are taken from Shi et al. (2023) and Geng et al. (2024). All results, except for those of the deterministic method MTUPlace3, are averaged over 5 runs with different random seeds and reported as $mean \pm std$. The symbols '+', '−', and '≈' indicate the number of circuits where the method performs significantly better than, worse than, or comparable to EGPlace, based on the Wilcoxon rank-sum test at a 0.05 significance level. Runtime comparisons for WireMask-EA, EfficientPlace, and EGPlace are also provided to highlight the computational efficiency of our method. The best results are marked in **bold**.

| Method | adaptec1 | adaptec2 | adaptec3 | adaptec4 | bigblue1 | bigblue3 | +/ − / ≈ | Avg. Rank |
|---|---|---|---|---|---|---|---|---|
| SP-SA | $18.84 \pm 4.62$ | $117.36 \pm 8.73$ | $115.48 \pm 7.56$ | $120.03 \pm 4.25$ | $5.12 \pm 1.43$ | $164.70 \pm 19.55$ | 0/6/0 | 7.7 |
| NTUPlace3 | 26.62 | 321.17 | 328.44 | 462.93 | 22.85 | 455.53 | 0/6/0 | 10.2 |
| RePlace | $16.19 \pm 2.10$ | $153.26 \pm 29.01$ | $111.21 \pm 11.69$ | $37.64 \pm 1.05$ | $2.45 \pm 0.06$ | $119.84 \pm 34.43$ | 1/5/0 | 5.8 |
| DreamPlace | $15.81 \pm 1.64$ | $140.79 \pm 26.73$ | $121.94 \pm 25.05$ | $\mathbf{37.41 \pm 0.87}$ | $2.44 \pm 0.06$ | $107.19 \pm 29.91$ | 1/5/0 | 5.3 |
| GraphPlace | $30.10 \pm 2.98$ | $351.71 \pm 38.20$ | $358.18 \pm 13.95$ | $151.42 \pm 9.72$ | $10.58 \pm 1.29$ | $357.48 \pm 47.83$ | 0/6/0 | 9.8 |
| DeepPR | $19.91 \pm 2.13$ | $203.51 \pm 6.27$ | $347.16 \pm 4.32$ | $311.86 \pm 56.74$ | $23.33 \pm 3.65$ | $430.48 \pm 12.18$ | 0/6/0 | 10.0 |
| MaskPlace (3k) | $7.62 \pm 0.67$ | $75.16 \pm 4.97$ | $100.24 \pm 13.54$ | $87.99 \pm 3.25$ | $3.04 \pm 0.06$ | $90.04 \pm 4.83$ | 0/6/0 | 5.7 |
| Chipformer (2k) | $6.62 \pm 0.05$ | $67.10 \pm 5.46$ | $76.70 \pm 1.15$ | $68.80 \pm 1.59$ | $2.95 \pm 0.04$ | $72.92 \pm 2.56$ | 0/6/0 | 4.7 |
| WireMask-EA (1k) | $6.15 \pm 0.05$ (3.16h) | $64.38 \pm 4.43$ (1.96h) | $58.18 \pm 1.04$ (1.42h) | $59.52 \pm 1.71$ (2.44h) | $2.15 \pm 0.01$ (1.02h) | $59.85 \pm 3.39$ (6.99h) | 2/3/1 | 3.0 |
| EfficientPlace (1k) | $5.94 \pm 0.04$ (0.84h) | $46.79 \pm 1.60$ (1.03h) | $\mathbf{56.35 \pm 0.99}$ (2.11h) | $58.47 \pm 1.61$ (4.50h) | $\mathbf{2.14 \pm 0.01}$ (1.20h) | $58.38 \pm 0.54$ (3.77h) | 2/3/1 | 2.0 |
| EGPlace (1k) | $\mathbf{5.85 \pm 0.08}$ (**0.32h**) | $\mathbf{37.39 \pm 1.58}$ (**0.76h**) | $61.09 \pm 1.00$ (**0.75h**) | $55.54 \pm 1.64$ (**1.18h**) | $2.24 \pm 0.03$ (**0.35h**) | $\mathbf{50.89 \pm 4.69}$ (**1.30h**) | | **1.8** |

(Cheng et al., 2022), MaskPlace (Lai et al., 2022), Chipformer (Lai et al., 2023), EfficientPlace (Geng et al., 2024b), the hybrid optimization method WireMask-EA (Shi et al., 2023), analytical based methods DreamPlace (Lin et al., 2019), NTUPlace3 (Chen et al., 2008), RePlace (Cheng et al., 2018) and simulated annealing based method SP-SA (Murata et al., 2002). All evaluations run on a standardized platform with an NVIDIA RTX 3090 Ti GPU and Intel Xeon Silver 4210R CPUs (2.40GHz). Our code is provided at https://github.com/dengji1/EAPlace.

**Macro Placement Results.** The macro placement results of EGPlace and the comparison methods on six benchmark circuits are summarized in Table D.2. For the two largest circuits, "bigblue2" and "bigblue4", only a subset of macros is considered to ensure computational feasibility, and the corresponding results are provided in Appendix D.2. Each placement method is executed with five different random seeds, and the mean and standard error of HPWL are reported. Among the six circuits, EGPlace achieves the best average ranking across all methods based on the Wilcoxon rank-sum test. In addition, as shown in Appendix D.2, EGPlace consistently outperforms all baseline methods on both bigblue2 and bigblue4. Across the entire ISPD2005 dataset, EGPlace surpasses the state-of-the-art EfficientPlace method, yielding a **9.3%** reduction in HPWL. EGPlace also demonstrates high computational efficiency, completing 1,000 iterations in an average of **0.81 hours**, which is **2.8×** and **7.8×** faster than EfficientPlace and WireMask-EA, respectively. These results highlight the effectiveness of our guided mutation strategy in efficiently exploring the solution space, particularly for large-scale circuits such as "bigblue3" and "bigblue4". Additionally, we have conducted further experiments on the ICCAD2015 benchmark and present the

results in Appendix D.5.

We further compare EGPlace with existing methods by analyzing the HPWL progression during the optimization process. The trend of the best HPWL over the number of iterations and runtime (in seconds) are plotted in Fig. 3 and Fig. 9 in the Appendix, respectively. The results demonstrate that EGPlace outperforms existing methods on 6 out of 8 benchmarks in terms of both sample and time efficiency.

**Congestion Results.** We investigate the congestion results of EGPlace, measuring congestion with the RUDY metric (Spindler & Johannes, 2007). First, we set $\lambda_1$ in Equation (1) to 0, focusing solely on optimizing HPWL and overlap. The HPWL and RUDY values for the 8 benchmarks are presented in Table 8 in the Appendix. EGPlace achieves smaller HPWL and RUDY on 6 out of 8 datasets, aligning with the findings in Shi et al. (2023), where shorter HPWL may lead to smaller congestion.

Moreover, we allow users to specify $\lambda_1$ to generate layouts with desired congestion levels. As "adaptec2" achieves the best results among benchmarks containing a full set of modules, we conduct congestion experiments, along with ablation and parameter analysis, on this benchmark to demonstrate the advantages of our method and the functionality of its components. The RUDY and HPWL values of layouts generated by EGPlace with $\lambda_1$ ranging from [0, 2.5] on "adaptec2" are shown in Fig. 4. We observe that increasing $\lambda_1$ leads to reduced congestion and increased HPWL in general. However, HPWL also influences congestion, as smaller HPWL values tend to result in lower global congestion and higher local congestion, as shown in previous work (Wang et al., 2000). This might explain the fluctuations in the HPWL and RUDY curves.

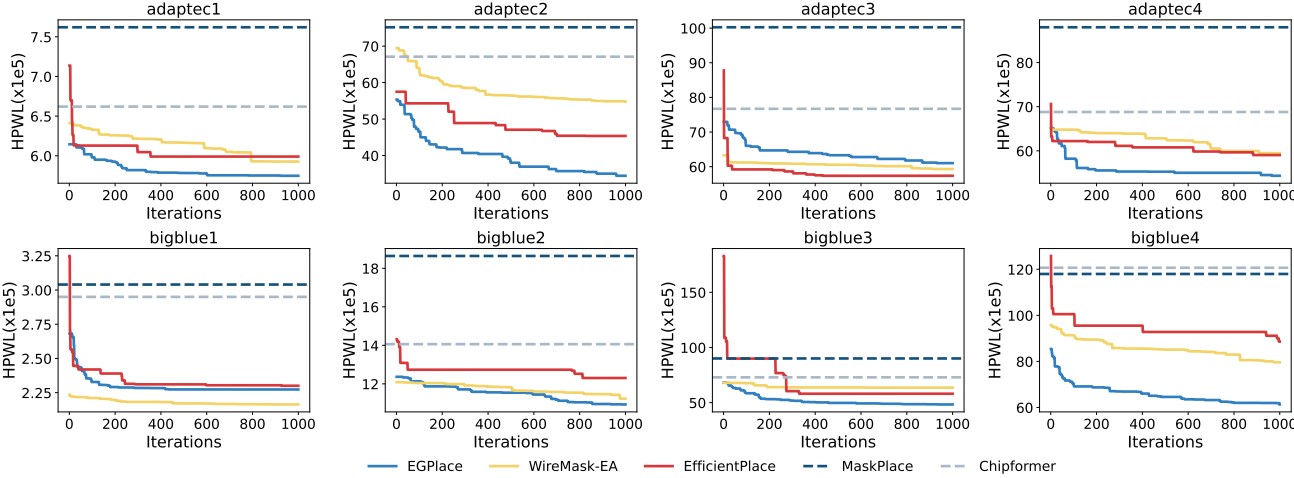

Figure 3. **Comparison of HPWL Trend Over the Number of Iterations.** We run the released code of WireMask-EA and EfficientPlace to generate the results.

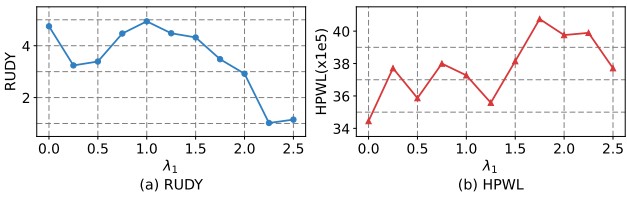

Figure 4. **RUDY and HPWL of Layouts Generated by EGPlace with Different $\lambda_1$ Values.** We run EGPlace on the "adaptec2" benchmark and present the RUDY and HPWL results in subplots (a) and (b), respectively.

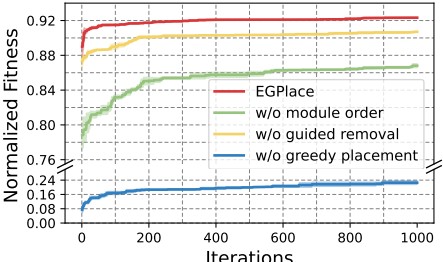

Figure 5. **Ablation Studies**. We test EGPlace on different configurations using the "adaptec2" benchmark and plot the average normalized fitness values of all individuals in the population at each iteration. The shaded area represents the standard error of the fitness values across the individuals.

**Mixed-size Placement Results.** Modern integrated circuits contain both macro blocks and numerous smaller standard cells, making mixed-size placement a critical real-world requirement. In this experiment, we demonstrate the effectiveness of our method in mixed-size placement by using EGPlace for macro placement and DreamPlace (Lin et al., 2019) for standard cell placement. The mixed-size placement results is provided in Table D.7 in the Appendix. The results show that our macro placement approach positively impacts the mixed-size placement quality.

**Ablation Studies.** We conduct ablation studies to investigate our choices' impact on performance by testing the performance on "adaptec2" with different configurations: (1) **w/o greedy reposition**: We randomly choose a valid location to position the modules, rather than placing them in the location that minimizes the increase in HPWL; (2) **w/o module order**: In each step, we randomly select a module for placement, rather than ordering the modules based on their sizes and connections; (3) **w/o guided removal**: We select modules for removal with uniform probability, instead of

selecting them with probability proportional to their impact on the layout quality; (4) **EGPlace**: The full configuration of our method. Performance evaluation uses normalized fitness values tracked across optimization iterations, with results presented in Fig. 5. The complete EGPlace system consistently outperforms all ablated variants. The greedy repositioning strategy proves particularly important, with its removal causing the most substantial performance degradation. The quality-guided module removal mechanism also provides significant benefits, improving both convergence speed and final solution quality. The module ordering strategy contributes measurably to overall system performance.

**Parameter Sensitivity Analysis.** We analyze the impact of population size and the module removal ratio on the mutation operator. The fitness value and HPWL of the layouts on "adaptec2" generated by EGPlace with population sizes between [1, 9] over 1,000 iterations are shown in Fig. 6. The

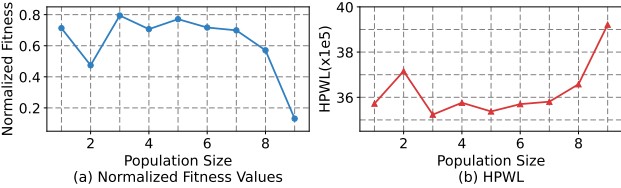

*Figure 6.* **Layout Quality of EGPlace with Different Population Sizes.** We run EGPlace for 1,000 iterations on "adaptec2" and plot the normalized fitness values and HPWL of the resulting layouts in subfigures (a) and (b), respectively.

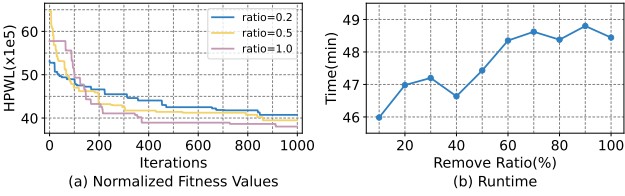

*Figure 7.* **Layout Quality of EGPlace with Different Removal.** We run EGPlace for 1,000 iterations on "adaptec2" and plot the normalized fitness values and runtime in subfigures (a) and (b).

results indicate that a population size that is too small (e.g., 1 or 2) may lead to under-exploration of the solution space, resulting in lower layout quality. Conversely, a population size that is too large (e.g., 8 or 9) can reduce the average number of mutations applied to each layout within the specified total number of iterations, thus lowering the quality. The optimal population size lies between 3 and 6.

We test different ratios (ranging from 10% to 100%) of modules for removal and repositioning in EGPlace, and run our method for 1,000 iterations on "adaptec2". The HPWL and runtime are shown in Fig. 7. We observe that a higher removal ratio leads to a slightly smaller HPWL, but also increases the runtime.

## 7. Conclusion and Future Work

This paper presents EGPlace, a evolutionary search placement method with a novel greedy repositioning guided mutation operator for macro placement optimization in integrated circuit design. In order to overcome the challenges in hybrid methods, EGPlace performs greedy repositioning guided mutation to achieve good performance, and integrates layout reconstruction into the mutation operator to improve efficiency. Experimental evaluation on industry-standard benchmarks show that EGPlace achieves consistent improvements in layout quality while significantly reducing computational requirements compared to state-of-the-art methods.

**Limitations and Future Research.** While the proposed EGPlace method demonstrates strong performance in macro placement and achieves competitive results on surrogate metrics such as HPWL and RUDY, the current design does not sufficiently account for mixed-size placement and does not directly target final Power, Performance, and Area (PPA) design objectives. Extending EGPlace to better support mixed-size placement and align more closely with PPA-driven optimization represents an important direction for future work. Recent advances, including MaskRegulate (Xue et al., 2024), which introduces a Regulate Mask to guide mixed-size placement, and LaMPlace (Geng et al., 2024a), which leverages a learned L-Mask to reflect PPA objectives, offer promising techniques. Integrating those masks into EGPlace holds potential for further improving placement quality and enhancing real-world applicability.

## Acknowledgements

This work is supported by NSFC (No. 62272008). We would like to thank the reviewers for their valuable suggestions.

## Impact Statement

The increasing complexity of modern integrated circuits demands more sophisticated electronic design automation (EDA) tools, particularly for chip placement optimization. Our work advances this critical area by introducing EGPlace, an efficient placement method that significantly improves both placement quality and computational efficiency. The method's ability to reduce wirelength by 6.7% to accelerate optimization by 7.8× compared to state-of-the-art methods demonstrates its practical value for industrial chip design.

Beyond immediate performance gains, EGPlace's unified optimization approach establishes new directions for scalable placement tools. These innovations become particularly valuable as the semiconductor industry moves toward more complex designs involving 3D integration and heterogeneous architectures. The method's flexible congestion control also enables designers to generate application-specific layouts, supporting diverse requirements across different market segments from mobile computing to high-performance systems.

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

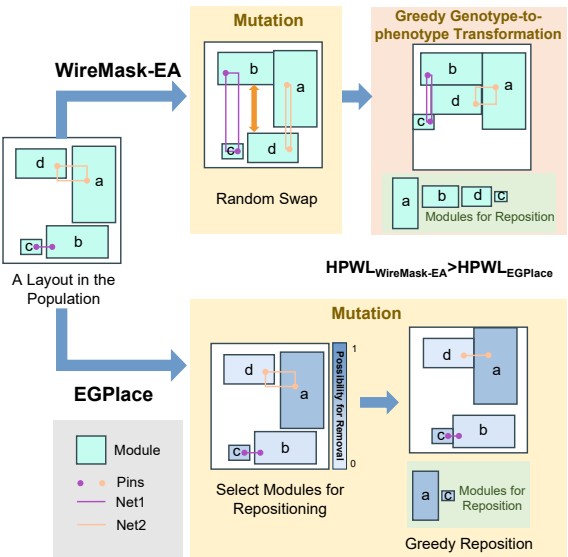

*Figure 8.* **Comparison between EGPlace and WireMask-EA.**

# Appendix

The Appendix provides a comprehensive technical exploration of EGPlace, offering detailed insights into its algorithmic design, computational strategies, and experimental validations. It elaborates on further technical details that underpin the proposed macro placement approach.

The supplementary materials cover four primary areas: a comparative analysis with WireMask-EA, an in-depth examination of fundamental placement concepts and metrics, a rigorous assessment of the computational complexity of the proposed algorithms, and an extensive presentation of experimental results across multiple benchmarks.

By presenting these technical details, the appendix offers researchers and practitioners a thorough understanding of the methodological innovations and performance characteristics of EGPlace, which extends beyond the summarized findings in the main manuscript. This supplementary content not only substantiates the research claims but also provides a transparent view into the algorithmic intricacies that enable the framework's superior performance in integrated circuit macro placement optimization.

## A. Comparison between EGPlace and WireMask-EA

We analyze the differences between our EGPlace and the state-of-the-art combined method WireMask-EA (Shi et al., 2023), both of which utilize evolutionary search as the backbone. Fig. 8 illustrates the layout adjustment process for a layout selected from the population in both methods. The main improvement of our method compared to WireMask-EA lies in two aspects: firstly, WireMask-EA utilizes random swap mutations, where poorly selected module pairs may lead to low-quality solutions. In contrast, EGPlace selects and repositions modules based on their impact on layout quality, increasing the likelihood of producing higher-quality mutated layouts. Secondly, WireMask-EA requires a separate step to transform the genotype into a phenotype. This step positions all modules in locations that minimize HPWL increment and are closest to those in the genotype. In comparison, EGPlace incorporates greedy placement directly into the mutation operator, requiring only the repositioning of selected modules, which significantly improves efficiency.

## B. Basic Concepts and Placement Metrics

The key concepts in chip placement are described as bellow.

- Chip: A chip is a rectangular region, often divided into $B \times B$ bins to facilitate metric calculations. A larger $B$ value provides higher resolution for module placement.

*Table 2.* Notations for Complexity Analysis

| Symbols | Meanings |
|---|---|
| $B$ | The number of bins in each row or column of the placement region |
| $m$ | A module |
| $M$ | The set of modules |
| $P$ | The set of pins |
| $P_m$ | The set of pins on module $m$ |
| $N$ | The set of nets |
| $f_{greedy}$ | The greedy placement process described in Algorithm 1 |
| $f_{select}$ | The module selection process outlined in Section 5.1 |
| $rm\%$ | The ratio of modules that undergo repositioning |

- Module: Modules are rectangles with fixed sizes and orientations. Macro cells are primarily functional modules, such as Dynamic Random Access Memory (DRAM) and caches. Standard cells are smaller in size, mainly consisting of logic gates.

- Pin: A pin is an input/output interface on a circuit module. It is fixed relative to the module and connects the module to others through nets.

- Port: A port is an input/output interface located at fixed positions on the chip. It connects to other pins or ports through nets.

- Net: A net consists of a set of connected pins. In digital circuits, pins in a net receive the same 0/1 signal.

Metrics measuring the layout quality include Half Perimeter Wire Length (HPWL) (Chen et al., 2006), Rectangular Uniform wire DensitY (RUDY) (Spindler & Johannes, 2007) and overlap rate (Lai et al., 2022). HPWL is an commonly used estimation of the routing wirelength. A short HPWL implies low routing resources, low delay, and low energy consumption. In HPWL calculation, the net bounding box is defined as the smallest rectangle enclosing all the pins of the net. The total HPWL of a layout is the sum of the HPWL of all nets, as shown in Equation (7) (Chen et al., 2006), where $E$ is a net containing pin $p$.

$$HPWL = \sum_E (\max_{p \in E} p_x - \min_{p \in E} p_x + \max_{p \in E} p_y - \min_{p \in E} p_y) \tag{7}$$

RUDY measures the congestion of the layout. Higher congestion may result in increased signal transmission delay, larger power consumption, and lower performance. The value of RUDY for each bin of the placement region is defined as the sum of the reciprocal of the height and width of the bounding boxes of all nets covering it. The value of the layout's RUDY is the maximum of the RUDY values for all bins.

The overlap rate is defined as the ratio between the modules' overlapping area and the total chip area.

## C. Implement Details and Complexity Analysis

### C.1. Efficient Computation of Module Overlaps

We provide a detailed explanation of the efficient computation of module overlap along the $x$-direction in Algorithm 2. This algorithm computes the overlap in $O(B)$, where $B$ denotes the number of bins per row in the placement region. The total overlap is obtained by combining the overlap lengths along the $x$- and $y$-directions between each pair of modules.

### C.2. Workflow of the Evolutionary Search Backbone and Overall Complexity Analysis

The overall workflow of the evolutionary search backbone is outlined in Algorithm 3. The notations for complexity analysis is demonstrated in Table 2. The time complexity is analyzed as follows: it takes $C(n_0 f_{greedy})$ to initialize the population. In each iteration, a layout is selected, and a mutation operator is applied to the layout. The mutation operator consists of two phases: module selection and repositioning, leading to a time complexity of $C(f_{select} + rm\% \cdot f_{greedy})$. Therefore, the

---

**Algorithm 2** Calculation for Overlap between Two Modules in the x-direction

---

**Input:** The placement area divided into $B \times B$ bins, a module $m$ to be placed, and module $n$ with its bottom right corner placed at $(n.x, n.y)$.

**Output:** The overlap between module $m$ and $n$ in the $x$-direction, denoted as $overlap_x$.

  1: $overlap_x[B][B] \leftarrow 0$
  2: $col \leftarrow max(0, n.x - m.width)$
  3: **while** $col < min(n.x, n.x + n.width - m.width)$ **do**
  4:    $overlap_x[col, :] \leftarrow col + m.width - n.x$
  5:    $col \leftarrow col + 1$
  6: **end while**
  7: **while** $col < n.x + n.width - m.width$ **do**
  8:    $overlap_x[col, :] \leftarrow m.width$
  9:    $col \leftarrow col + 1$
10: **end while**
11: **while** $col < min(B, n.x + n.width)$ **do**
12:    $overlap_x[col, :] \leftarrow n.x + n.width - col$
13:    $col \leftarrow col + 1$
14: **end while**

**Return:** $overlap_x$

---

overall time complexity is $O(n_0 C(f_{greedy}) + I_{max}(C(f_{select} + rm\% \cdot f_{greedy})))$. A detailed analysis of the complexity of $f_{select}$ and $f_{greedy}$ is provided in the following paragraphs.

The module selection process, $f_{select}$, involves calculating the score for each module based on Equation (3), which consists of three terms. The first term, $wirelen_m$, computes the sum of the Manhattan distances between each pin on the module and the center of its net bounding box. The net bounding box positions are recorded during the greedy placement, so the center coordinate of each bounding box can be calculated in $O(1)$ time. Thus, the calculation of $wirelen_m$ for all modules requires $O(1)$ time for each pin, resulting in a total time complexity of $O(|P|)$. For the second term, $cong_m$, we first compute the congestion of each bin in $O(|N|)$ time using the recorded net boundaries. Then, the RUDY value for every net and the layout $L$ is computed by aggregating the values of the corresponding bins, resulting in a complexity of $O(|N|)$. When calculating $cong_m$ for each module, we check whether the congestion of the net associated with each module exceeds the threshold with in $O(1)$ time. The total time for this is $O(|P|)$. Therefore, the overall time complexity for this term is $O(|N| + |P|)$, which can be simplified to $O(|P|)$ as $|N| < |P|$. The third term, $overlap_m$, can be calculated in $O(|M|)$ by aggregating the module coverage for each bin recorded during greedy repositioning. The overall module selection process results in a complexity of $O(|P| + |M|)$.

$f_{greedy}$ involves the computation of the wire mask, bound mask and overlap mask before the placement of each module. We adopt the idea in MaskPlace(Lai et al., 2022) to optimize wire mask calculations, which reduces the complexity to $O(BP_m)$ by caching the boundaries of nets. The bound mask computation can be easily simplified to $O(B)$ by directly verifying the grids near to the chip boundaries. Suppose $m$ is the $i$-th module to be placed, the overlap mask computation for $m$ results in a time complexity of $O(Bi)$. Therefore, the total computation time before placing one module result in $O((|P| + |M|^2)B)$, and the overall computation time of $f_{greedy}$ is $O((|P| + |M|^2)|M|B)$.

## D. Experimental Details and Additional Results

### D.1. Dataset and Experimental Details

The statistics of the ISPD2005 circuit benchmark (Nam et al., 2005) and the Ariane RISC-V CPU design benchmark (Zaruba & Benini, 2019) are demonstrated in Table 3, where "Macros(to place)" stands for the number of macros that is chosen for placement by EGPlace. We have also conducted further macro placement experiments on the modern ICCAD2015 benchmark (Kim et al., 2015), and present the statistics in Table 4.

The hyperparameters used in the experiments are configured as follows: the population size is set to 5; the number of bins is $224 \times 224$; the removal ratio in the mutation operator is set to 50%; the ratio $r\%$ in module score computation is set to 98; $\lambda_2$ in the objective function is set to 0.1 As congestion is not considered in previous methods, we set the congestion weight

---

**Algorithm 3** Workflow of Evolutionary Search Backbone

---

**Input:** A placement region divided into $B \times B$ bins, a netlist $N$ that defines the features and connectivity of the modules, population size $n$, number of initial layouts $n_0 \leq n$, hyperparameters $\lambda_1$ and $\lambda_2$ to balance the metric weights in the objective function, and the maximum number of iterations $I_{max}$.
**Output:** An optimized layout of circuit modules.

1:  Initialize an empty layout population $P$ of size $n$
2:  $i \leftarrow 0$
3:  **while** $i < n_0$ **do**
4:      Generate an initial layout $L$ using Algorithm 1 and add it to $P$
5:  **end while**
6:  $i \leftarrow 0$
7:  **while** $i < I_{max}$ **do**
8:      Select a layout $L$ from $P$ with probability given by Equation (2)
9:      Select a set of modules $M$ for repositioning from L according to Section 5.1.
10:     Remove $M$ from $L$
11:     Obtain the mutated layout $L_1$ by repositioning $M$ according to Algorithm 1.
12:     $P \leftarrow P \cup \{L_1\}$
13:     **while** $|P| > k$ **do**
14:         Remove the layout in $P$ with the lowest fitness
15:     **end while**
16:     $i \leftarrow i + 1$
17: **end while**
**Return:** The layout in $P$ with the highest fitness

---

*Table 3.* Statistics of the ISPD2005 Circuit Benchmark

| Circuit | Macros | Macros(to place) | Standard Cells | Nets | Pins | Ports | Area Util%) |
|---|---|---|---|---|---|---|---|
| adaptec1 | 543 | 543 | 210904 | 221142 | 944063 | 0 | 55.62 |
| adaptec2 | 566 | 566 | 254457 | 266009 | 1069482 | 0 | 74.46 |
| adaptec3 | 723 | 723 | 450927 | 466758 | 1875039 | 0 | 61.51 |
| adaptec4 | 1329 | 1329 | 494716 | 515951 | 1912420 | 0 | 48.62 |
| bigblue1 | 560 | 560 | 277604 | 284479 | 1144691 | 0 | 31.58 |
| bigblue2 | 23084 | 1024 | 534782 | 577235 | 2122282 | 0 | 32.43 |
| bigblue3 | 1293 | 1293 | 1095519 | 1123170 | 3833218 | 0 | 66.81 |
| bigblue4 | 8170 | 1024 | 2169183 | 2229886 | 8900078 | 0 | 35.68 |
| ariane | 932 | 932 | 0 | 12404 | 22802 | 1231 | 78.39 |

$\lambda_1$ to 0 in the objective function for fair comparison. Notably, $\lambda_1$ and $\lambda_2$ can be adjusted by the user to generate desired layouts.

### D.2. Results on the "bigblue3" and "bigblue4" Circuits

To ensure computational feasibility on the large-scale circuits "bigblue2" and "bigblue4", we select the first 1,024 macros based on our module ordering strategy described in Section 5.2. The results are presented in Table **??**. The results show that EGPlace achieves the lowest HPWL on both benchmarks, outperforming the recent EfficientPlace method by **8.5%** and **28.7%**, respectively. In addition, EGPlace demonstrates substantially lower runtime compared to both EfficientPlace and WireMask-EA.

### D.3. Time Efficiency

Figure 9 demonstrates the HPWL trend over runtime in EGPlace and the comparation methods. The results show that EGPlace achieves better results on 6 out of 8 benchmarks.

*Table 4.* Statistics of the ICCAD2015 Circuit Benchmark

| Circuit | Macros | Macros (to place) | Stdandard Cells | Nets | Pins |
|---|---|---|---|---|---|
| superblue1 | 56898 | 512 | 1159346 | 1215710 | 3767494 |
| superblue3 | 58970 | 512 | 1160765 | 1224979 | 3905321 |
| superblue4 | 45289 | 512 | 756979 | 802513 | 2497940 |
| superblue5 | 76676 | 512 | 1014341 | 1100825 | 3246878 |
| superblue7 | 72256 | 512 | 1865884 | 1933945 | 6372094 |
| superblue10 | 101837 | 512 | 1786523 | 1898119 | 5560506 |
| superblue16 | 4868 | 512 | 981140 | 999902 | 3013268 |
| superblue18 | 27099 | 512 | 744947 | 771542 | 2559143 |

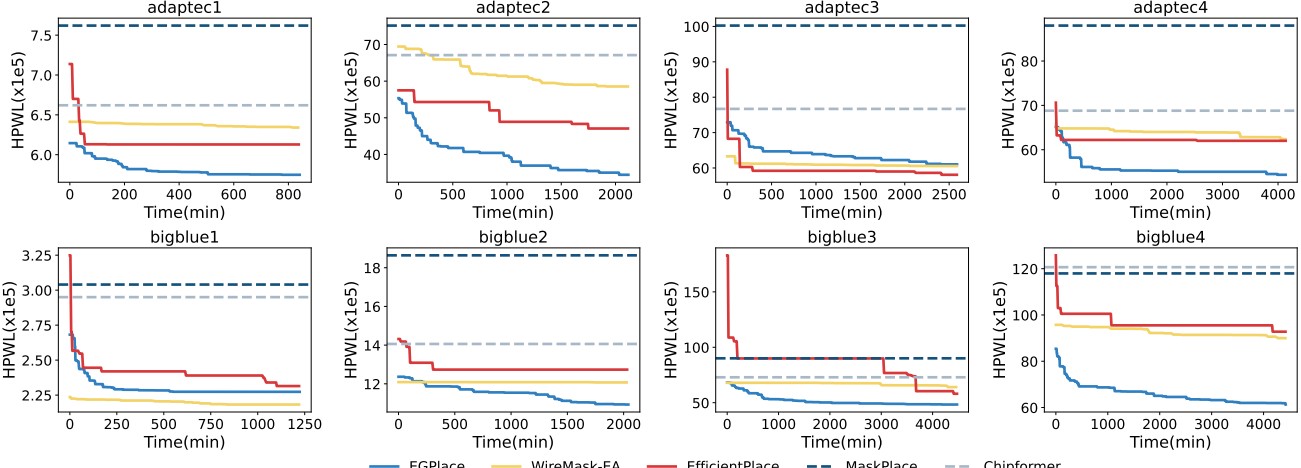

*Figure 9.* **Comparison of HPWL Trend Over the Runtime(s).**

*Table 5.* **HPWL Values($\times 10^5$) Obtained from 5 Macro Placement Methods on "bigblue2" and "bigblue4" circuits.** We select 1,024 modules as macros for "bigblue2" and "bigblue4" and run the comparison methods with their default hyperparameter configurations. We report the runtime of WireMask-EA, EfficientPlace, and EGPlace to highlight the efficiency of our method. The best results are marked in **bold**.

| Benchmark | MaskPlace (3k) | Chipformer (2k) | WireMask-EA (1k) | EfficientPlace (1k) | EGPlace (1k) |
|---|---|---|---|---|---|
| bigblue2 | $18.64 \pm 0.63$ | $14.06 \pm 0.47$ | $11.35 \pm 0.15$ (19.83h) | $12.20 \pm 0.29$ (2.00h) | **$11.16 \pm 0.47$ (0.58h)** |
| bigblue4 | $117.96 \pm 5.62$ | $120.66 \pm 8.03$ | $82.96 \pm 2.32$ (13.87h) | $86.86 \pm 3.41$ (2.85h) | **$61.90 \pm 2.73$ (1.26h)** |

*Table 6.* **HPWL and Overlap Results on the "ariane" Benchmark.** We obtain the result of MaskPlace and DreamPlace by running their released code. We modify the code of EfficientPlace to run on the "ariane" benchmark and include ports in HPWL calculation for fair comparison.

| Method | MaskPlace | DreamPlace | EfficientPlace | EGPlace |
|---|---|---|---|---|
| Overlap rate (%) | 3.27 | LG failed | 3.37 | **1.24** |
| HPWL ($\times 10^5$) | 14.63 | LG failed | 12.47 | **7.91** |

## D.4. Results on the "ariane" Benchmark

We conduct experiments on the "ariane" benchmark to evaluate the effectiveness of the overlap mask described in Section 5.2. As shown in Table 3, the "ariane" benchmark consists of macro modules covering 78.39% of the chip area, so it is difficult for chip placement methods to generate layouts without overlap. Our goal is to produce layouts with minimal HPWL and as little overlap as possible.

We compare EGPlace with the state-of-the-art analytical method DreamPlace and RL-based methods MaskPlace and EfficientPlace. MaskPlace and EfficientPlace use position masks with binary values to indicate whether module placement causes overlap, whereas our method employs an overlap mask that calculates the exact overlap value and combines it with wire mask for greedy placement. Table 6 presents the results on the "ariane" benchmark. The result shows that DreamPlace results in significant overlap, leading to legalization failure. EGPlace achieves a significantly lower overlap rate than MaskPlace and EfficientPlace, demonstrating the effectiveness of the overlap mask in reducing module overlap.

## D.5. Results on ICCAD2015 Benchmark

We further conduct experiments on the ICCAD2015 benchmark. The placement files in LEF/DEF format are converted into bookshelf format using ChiPBench (Wang et al., 2024), and the 256 largest modules in each circuit are selected for macro placement. The results are presented in Table 7. EGPlace achieves HPWL reductions of 49.5%, 17.4%, and 7.4% compared to the baselines DreamPlace, WireMask-EA, and EfficientPlace, respectively. These results demonstrate the effectiveness of our method in macro placement.

*Table 7.* **Comparison of HPWL Results($\times 10^8$) on the ICCAD2015 benchmark for different methods.** We run WireMask-EA and EfficientPlace with the number of grids set to 224 and 512, respectively, while other settings remain as default. We highlight the best results in **bold** and underline the second best results.

| Method | superblue1 | superblue3 | superblue4 | superblue5 | superblue7 | superblue10 | superblue16 | superblue18 |
|---|---|---|---|---|---|---|---|---|
| DreamPlace | 2.53 | 15.19 | 3.44 | 21.36 | 5.09 | 12.99 | 2.66 | 1.02 |
| WireMask-EA(1k) | 1.37 | 4.40 | 2.11 | 11.00 | **2.86** | 1.18 | 2.85 | 1.46 |
| EfficientPlace(1k) | **1.26** | 3.81 | 1.99 | 9.70 | **2.86** | **0.93** | 2.79 | 1.12 |
| EGPlace(1k) | 1.31 | **3.22** | **1.91** | **8.62** | 2.90 | 1.00 | **2.03** | **0.96** |

## D.6. Congestion Results

A comparison of RUDY and HPWL between EGPlace and the comparative methods is illustrated in Fig. 8. We set the weight of RUDY in the objective function $\lambda_1$ to 0, and demonstrate the congestion levels of layouts obtained by optimizing solely HPWL and overlap. The results indicate that EGPlace achieves lower congestion on 6 out of 8 benchmarks, aligning with findings in WireMask-EA (Shi et al., 2023), which suggest that a lower HPWL may contribute to reduced congestion. Furthermore, by adjusting $\lambda_1$, we can generate layouts that meet specific congestion requirements.

*Table 8.* **Comparasion on HPWL($\times 10^5$) and Congestion Results.** We obtain the results of the comparison methods by running their publicly released code. For a fair comparison, the RUDY values are calculated on the placement region divided into $224 \times 224$ bins. Following previous methods (Shi et al., 2023; Geng et al., 2024b), we standardize the RUDY value of EGPlace to 1. We highlight the best results in **bold**.

| Benchmarks | adaptec1 | | adaptec2 | | adaptec3 | | adaptec4 | | bigblue1 | | bigblue2 | | bigblue3 | | bigblue4 | |
| Metrics | HPWL | Cong. | HPWL | Cong. | HPWL | Cong. | HPWL | Cong. | HPWL | Cong. | HPWL | Cong. | HPWL | Cong. | HPWL | Cong. |
|---|---|---|---|---|---|---|---|---|---|---|---|---|---|---|---|---|
| MaskPlace | 7.09 | 1.51 | 78.57 | 3.23 | 118.32 | **1.00** | 91.22 | 3.80 | 2.67 | **1.00** | 17.49 | 2.17 | 96.91 | 2.56 | 112.87 | 0.59 |
| WireMask-EA | 5.93 | **0.97** | 54.76 | 1.23 | 59.31 | 1.01 | 59.47 | 3.10 | **2.16** | 2.63 | 11.23 | 1.34 | 66.05 | 1.56 | 79.59 | 0.37 |
| EfficientPlace | 5.99 | 1.05 | 45.37 | 2.45 | **57.37** | 1.69 | 59.04 | 2.20 | 2.30 | 1.18 | 12.30 | **1.00** | 58.08 | 1.22 | 88.60 | **0.26** |
| EGPlace | **5.75** | 1.00 | **37.99** | **1.00** | 63.05 | **1.00** | **56.09** | **1.00** | 2.23 | **1.00** | 10.49 | **1.00** | 50.50 | **1.00** | **58.96** | 1.00 |

## D.7. Mixed-size Placement Results

We perform mixed-size placement by adopting the two-phase approach in EfficientPlace (Geng et al., 2024b): (1) In the first phase, we fix all macros placed by EGPlace, and utilize the global placement step of DreamPlace to position the standard cells, generating coarse layouts. (2) In the second phase, we set all modules movable and carry out a comprehensive mixed-size placement process using DreamPlace, which includes the global placement step, legalization, and the detailed placement step. The mixed-size placement results are shown in Table D.7. The results of the comparison methods are taken from the EfficientPlace paper (Geng et al., 2024b) which adopts a previous version of DreamPlace for mixed-size placement. However, since recently DreamPlace 4.1.0 (Chen et al., 2023) has shown significant improvements over its previous version in mixed-size placement, we also conduct experiments using DreamPlace 4.1.0 and reported the results. The results show that layouts generated by EGPlace followed by DreamPlace 4.0 achieve an average HPWL reduction of 6.5% across eight circuits compared to EfficientPlace, demonstrating that good quality macro placement results serve as good initial solutions for mixed-size placement. We also find that integrating EGPlace with DreamPlace 4.1.0 leads to substantial improvements over the previous version of DreamPlace, however, these results remain slightly inferior to those achieved by DreamPlace 4.1.0 alone. As this work primarily focuses on macro placement, we leave further enhancements in mixed-size placement for future work.

*Table 9.* **Comparison on HPWL($\times 10^7$) for Mixed-size Placement on Different Datasets.** We conduct further experiments on DreamPlace 4.1.0 as they achieve significant improvement in placement quality than the previous versions. The results for comparison methods is are from the paper of EfficientPlace (Geng et al., 2024b). We mark the best results in **bold** and underline the second best results.

| Method | adaptec1 | adaptec2 | adaptec3 | adaptec4 | bigblue1 | bigblue2 | bigblue3 | bigblue4 |
|---|---|---|---|---|---|---|---|---|
| DreamPlace | 11.10 ± 1.31 | 13.84 ± 1.74 | 17.03 ± 0.99 | 24.37 ± 1.13 | 10.06 ± 0.28 | \ | 36.51 ± 0.56 | 175.86 ± 2.23 |
| DreamPlace 4.1.0 | 6.85 ± 0.24 | 8.50 ± 0.20 | **13.23 ± 0.15** | **13.15 ± 0.22** | **8.17 ± 0.01** | 9.89 ± 0.24 | 28.52 ± 0.46 | 63.34 ± 0.55 |
| MaskPlace+DreamPlace | 10.86 ± 0.01 | 12.98 ± 0.58 | 26.14 ± 0.07 | 23.52 ± 0.01 | 10.64 ± 0.01 | \ | 54.98 ± 1.06 | \ |
| WireMask-EA+DreamPlace | 8.93 ± 0.01 | 9.20 ± 0.05 | 21.72 ± 0.01 | 20.51 ± 0.01 | 10.35 ± 0.02 | 14.88 ± 0.01 | 42.52 ± 0.11 | 171.23 ± 0.48 |
| EfficientPlace+DreamPlace | 7.20 ± 0.12 | 9.20 ± 0.61 | 16.49 ± 1.07 | 14.70 ± 0.25 | 8.67 ± 0.10 | 9.98 ± 0.02 | **28.48 ± 0.96** | 125.02 ± 0.02 |
| EGPlace+DreamPlace 4.0 | 7.53 ± 0.11 | 9.06 ± 0.36 | 14.15 ± 0.19 | 15.69 ± 0.09 | 8.99 ± 0.06 | 9.63 ± 0.03 | 29.08 ± 0.23 | **62.32 ± 0.32** |
| EGPlace+DreamPlace 4.1.0 | **6.70 ± 0.36** | **8.02 ± 0.22** | 13.84 ± 0.88 | 13.44 ± 0.03 | 8.38 ± 0.08 | **9.57 ± 0.03** | 32.01 ± 0.32 | 66.63 ± 0.07 |

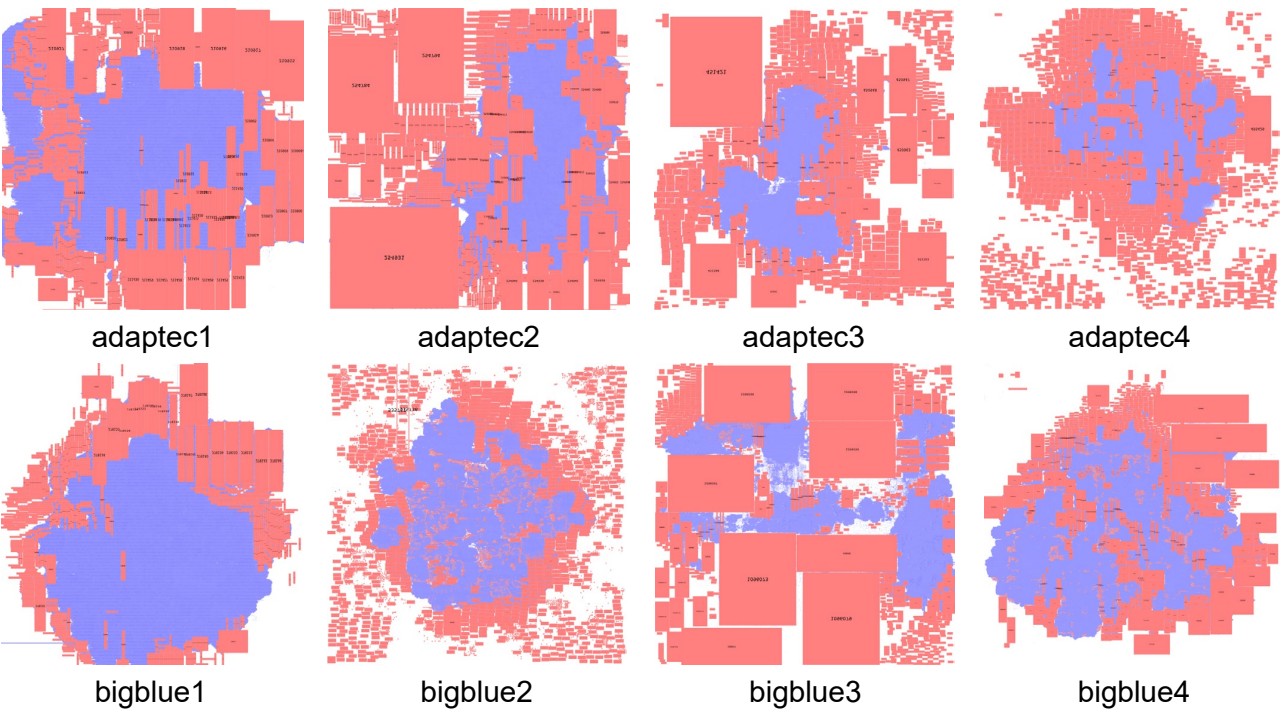

*Figure 10.* **Visualization of Mixed-size Placement Results on the ISPD 2005 Dataset.** Macros are marked in red, while standard cells are marked in blue.

