# OpenReview forum: "EGPlace: An Efficient Macro Placement Method via Evolutionary Search with Greedy Repositioning Guided Mutation"
_ICML.cc/2025/Conference — ICML 2025 poster_

### Official Review · Reviewer_iifF · 2025-02-24

**Overall Recommendation:** 2

**Summary:**

This paper proposes EGPlace, an evolutionary search-based approach for macro placement. It incorporates the wirelength, congestion, and overlap into its score computation. It achieves better HPWL and faster speed than previous RL-based approaches.

## update after rebuttal
After carefully reviewing the rebuttals and comments, I would like to maintain my current score.

**Claims And Evidence:**

Yes.

**Essential References Not Discussed:**

No.

**Experimental Designs Or Analyses:**

The validity of mixed-size placement is questionable. For the comparison between the proposed method EGPlace and analytical DREAMPlace, please see the "Reference Results for Macro Placement" part in https://github.com/limbo018/DREAMPlace. The results by DREAMPlace 4.1.0 are better than the EGPlace. Please discuss these results properly.

**Methods And Evaluation Criteria:**

Yes.

**Other Comments Or Suggestions:**

1. Macro Placement Results seem to be listed in Table 1 in the main paper, instead of Table 6 in the Appendix.
2. Typo: "araine" -> "ariane" in caption of Table 4.
3. In line 966, maybe it should be "it is 2.8x more efficient than EfficientPlace" instead of "EGPlace"?

**Other Strengths And Weaknesses:**

**Strengths**
1. An efficient approach to fast construct the overlap mask between current block and placed blocks. And it is capable to reflect the exact overlap value.

**Weaknesses**
1. The overlap rate of circuit ariane with approach MaskPlace, which is shown in Table 4 in the appendix, is 3.27%. However, in the paper of MaskPlace [1], the overlap of "ariane" shown in Table 3 with method "MaskPlace (hard constraint)", is 0.00. Please discuss the inconsistency between the overlap.
2. For the comparison between the proposed method EGPlace and analytical DREAMPlace, please see the "Reference Results for Macro Placement" [2] part in https://github.com/limbo018/DREAMPlace. The results by DREAMPlace 4.1.0 are better than the EGPlace. Please discuss these results properly. In the authors' submission, the results of DREAMPlace is much worse than other baselines, but according to the reference results shown in the github pages of DREAMPlace, EGPlace is worse than DREAMPlace. I think it is very significant to figure out this inconsistency since the gap between DREAMPlace in submission and github is very huge.
3. The overall technical contribution is limited, and the full pipeline is similar to the WireMask-BBO.
4. Other critical objectives are not taken into consideration, such as the post-routing wire length, timing metrics including WNS and TNS.


[1] Lai, Yao, Yao Mu, and Ping Luo. "Maskplace: Fast chip placement via reinforced visual representation learning." Advances in Neural Information Processing Systems 35 (2022): 24019-24030.

[2] https://github.com/limbo018/DREAMPlace

**Questions For Authors:**

1. In the calculation of $wirelen_m$ in Eq. 4, the authors claim that $e_p$ is the bounding box center of a net which contains pin $p$. If there are multiple pins containing pin $p$ simultaneously, how to address this case?
2. The same question for the $cong_m$ in Eq. 5, where $E_p$ is the RUDY value of the net containing pin $p$. If there are multiple pins containing pin $p$ simultaneously, how to address this case?
3. How to determine the sequence of module placement? It is better to directly show the sequence, instead of justing writing "larger and highly connected modules first".

**Relation To Broader Scientific Literature:**

The proposed method contributes to the application of ES algorithms into real and complex scene.

**Theoretical Claims:**

Yes. There are no issues about the correctness of any proofs for theoretical claims.

---

> ### Author Rebuttal · Authors · 2025-04-01
>
> Thank you for your constructive review and valuable suggestions! Below, we provide a detailed response to your questions and comments. If any of our responses fail to sufficiently address your concerns, please inform us, and we will promptly follow up.
>
> **The Results of MaskPlace on Circuit Ariane**
>
> Thanks for your feedback. We notice that the results in Table 4 of MaskPlace are normalized, so our results for MaskPlace and Chipformer are taken from Table 7 in the Chipformer paper. For further verification, we have run the released code of MaskPlace on the ariane benchmark for 1K iterations with grid size set to 224, and observed an overlap ratio of 3.33%. This suggests that on datasets with large module coverage areas, completely eliminating overlap can be challenging.
>
> **Difference Between EGPlace and WireMask-BBO**
>
> Although both WireMask-BBO and EGPlace employ a greedy strategy, there are several key differences between the two methods. 1) The evolutionary search variant in WireMask-BBO relies solely on random mutations to explore the search space, whereas EGPlace performs guided mutations to enhance layout quality and improve sampling efficiency. 2), EGPlace is significantly more efficient than WireMask-BBO. WireMask-BBO first applies layout mutation, followed by a greedy genotype-to-phenotype transformation that requires removing and repositioning all macros in the circuit. In contrast, EGPlace introduces a mutation operator that integrates layout adjustment and reconstruction phases, eliminating the computational overhead caused by full layout reconstruction after local modifications.
>
> From the perspective of experimental results, EGPlace achieves an average 10.79% improvement in HPWL and a 7.8× speedup in runtime (Table 1), which also verifies the new technical contribution of EGPlace.
>
> **Typo Correction**
>
> We sincerely apologize for the typographical errors. The macro placement results are listed in Table 1, where EGPlace is 2.8× more efficient than EfficientPlace. We will correct these errors in the revised version.
>
> **How to Address the Case where a Pin is Shared by Multiple Pins**
>
> A pin serves as an input/output interface of a module, and we do not think that pins contain each other. We assume that your question pertains to how the scores of modules are computed when a pin is associated with multiple nets. Based on this understanding, we provide our response bellow accordingly.
>
> In most cases, a pin belongs to only one net, as noted in Appendix A.1 of MaskPlace. In our implementation, we follow the same approach in competitors like MaskPlace, EfficientPlace, and WireMask-EA, by treating each input and output interface of a net in the .nets file as a separate pin. Consequently, if there exist a pin that appears in multiple nets, each occurrence is treated as an independent pin.
>
> We sincerely apologize if we have misunderstood your question. Please let us know if further clarification is needed.
>
> **How to Determine Module Placement Order**
>
> We determine the module placement order in the same manner as MaskPlace. We will include an algorithm for determining the module order into the appendix in the revised version. Specifically, the placement order of a module m is computed considering the size of the module (Size[m]), the number of nets (NodeNetNum[m]), and the number of direct neighbors placed previously (NeiScore[m]). Each time, we select one module and add it into the order list, and then update NeiScore[n] for each neighbor n of m.
>
> **Input**: A set of all modules M, a hash table Size storing the area of each module, a hash table NodeNetNum storing the number of nets each module belongs to, an adjacency matrix Adj where modules within the same net are considered as neighbors. α and β are hyperparameters set according to MaskPlace.
>
> **Output**: A sequence I that stores M in order.
>
> ```text
> 1. I ← ∅
> 2. NeiScore[m] ← 0 for all m ∈ M
> 3. While I ≠ M DO
> 4.   m ← Argmax(Size[m]*α + NodeNetNum[m]*β + NeiScore[m]) for m ∉ I
> 5.   I ← I ∪ {m}
> 6.   For Each n ∈ Adj[m] DO
> 7.     NeiScore[n] ← NeiScore[n] + 1
> 8. Return I
> ```
>
> **Performance of DreamPlace**
>
> Please refer to our feedback for Reviewer Nq5X.
>
> **Consideration of other PPA Objectives**
>
> EGPlace focuses on guided layout generation, including selecting layout candidates from the pool and choosing poorly-placed modules for reconstruction. Currently, the guidance comes from HPWL and other easily-computed metrics. It is not difficult for EGPlace to incorporate final metrics, such as those learned from the mask in LaMPlace (ICLR 2025).
>
> We have run OpenRoad to collect PPA metrics on layouts produced by EGPlace using the ariane133 dataset, without considering PPA metrics during the layout generation. As the process is slow, we will report the results when they become available.

---

### Official Review · Reviewer_nQ5x · 2025-03-10

**Overall Recommendation:** 3

**Summary:**

The manuscript proposes EGPlace, an innovative evolutionary optimization framework for macro placement in IC design, introducing a greedy repositioning-guided mutation operator and efficient mask computation algorithm. Experimental results show that EGPlace achieves significant improvements in wirelength reduction and computational speed compared to existing methods.

**Claims And Evidence:**

Yes.

**Essential References Not Discussed:**

NA

**Experimental Designs Or Analyses:**

The experiments cover most placement benchmarks and baselines. However, the experiment lacks a comparison with other analytic methods.

**Methods And Evaluation Criteria:**

- The main concern is that the method optimizes the macro's wire length. However, the results show that the mixed-size place method can also achieve the best HPWL results. However, the reason is not clearly explained in the paper.

- Also, there are works [1, 2] claim that optimizing wirelength does not directly optimize PPA. Therefore, whether it can ultimately improve chip performance remains open to discussion.

- The main contributions of this paper lie in the proposal of a novel mask computation method and improvements to the evolutionary algorithm. Its contributions to the ML community are relatively limited.

[1] Benchmarking End-To-End Performance of AI-Based Chip Placement Algorithms.

[2] Reinforcement Learning Policy as Macro Regulator Rather than Macro Placer.

**Other Comments Or Suggestions:**

NA

**Other Strengths And Weaknesses:**

While ICML is a prestigious conference in machine learning, its primary focus is on advancing the theoretical and practical aspects of machine learning algorithms and their applications. Evolutionary algorithms, unless tightly integrated with machine learning tasks (e.g., neural architecture search, hyperparameter optimization, or reinforcement learning), might not align well with the core interests of the ICML community. The paper may be more suitable for a domain conference.

**Questions For Authors:**

- How to get the results of Table 6 (mixed-size placement)? According to the Dreamplace repo (https://github.com/limbo018/DREAMPlace), the bigblue1 can get 8.62e7. However, all baselines are worse than Dreamplace.

- Table 5, MaskPlace get the 9.69 in HPWL for bigblue3. Is there a typo.

**Relation To Broader Scientific Literature:**

The contributions of this paper seem only applicable to macro placement tasks.

**Theoretical Claims:**

There are no theoretical claims.

---

> ### Author Rebuttal · Authors · 2025-04-01
>
> Thank you for your constructive review and valuable suggestions!
>
> **Regarding Mixed-size Placement**
>
> We conduct mixed-size placement following the same approach in Chipformer and EfficientPlace. We first fix all macros placed by EGPlace and apply the global placement step of DreamPlace to position the standard cells. We then set all modules as movable and perform a complete mixed-size placement process using DreamPlace, which includes global placement, legalization, and detailed placement.
>
> Note that in this paper, our primary focus is on macro placement, and the mixed-size placement process is not fully optimized, which may be further improved by incorporating standard cell clustering and considering additional surrogate metrics, as suggested in MacroRegulate and LaMPlace.
>
> **Regarding PPA Metrics**
>
> We fully agree that optimizing the final PPA metrics is the ultimate goal. However, we must note that incorporating final PPA metrics into online optimization is challenging, as obtaining them is time-consuming.
>
> EGPlace focuses on guided layout generation, including selecting layout candidates from the pool and choosing poorly-placed modules for reconstruction. Currently, the guidance comes from HPWL and other easily-computed metrics. It is not difficult for EGPlace to incorporate final metrics, such as those learned from the mask in LaMPlace (ICLR 2025).
>
> We perform placement using the current EGPlace method on the "ariane133" dataset and evaluate the PPA metrics through OpenRoad. As the process is slow, we will report the results when they become available.
>
> **Our Relationship with ML**
>
> Module placement is a well-known machine learning problem, and we have included ML baselines for comparison, showing that EGPlace achieves better performance with less running time. We believe that the audience at ICML would have sufficient interest in our work (if accepted).
>
> The current version of EGPlace serves as a strong starting point that can be further enhanced by machine learning algorithms. For instance, EGPlace can potentially be combined with the learned masks for PPA metrics improvement from LaMPlace.
>
> **Lack of Comparison with Analytic Methods**
>
> The results of analytical methods, including NTUPlace3, RePlace, and DREAMPlace, have already been compared with our primary competitors, MaskPlace and EfficientPlace, in their respective papers. Notably, MaskPlace and EfficientPlace have shown superior performance over these baselines. However, we plan to include results from additional competitors in the revised version to make our paper more comprehensive.
>
> **Only Applicable to Macro Placement**
>
> Similar to some baseline methods, including MaskPlace, EfficientPlace, and WireMask-EA, our work primarily focuses on macro placement, based on the observation that macro placement has a significant impact on layout metrics.
>
> Note that EGPlace supports mixed-size module placement. We have conducted mixed-size placement using DreamPlace and reported the results in Table 6 of our manuscript.
>
> **Performance of Dreamplace**
>
> We conduct mixed-size placement following the same approach in Chipformer and EfficientPlace. We first fix all macros placed by EGPlace and apply the global placement step of Dreamplace to position the standard cells. We then set all modules as movable and perform a complete mixed-size placement process using Dreamplace, which includes global placement, legalization, and detailed placement.
>
> For the results obtained by Dreamplace, we report the outcomes in Table 5 of the baseline method EfficientPlace for a fair comparison, where the results are close to those of Dreamplace 4.0.0. However, we observe that Dreamplace 4.1.0 on the mixed-size placement dataset are improved significantly, outperforming EGPlace.
>
> As our work primarily focuses on macro placement, we further evaluate the macro placement performance of EGPlace and Dreamplace 4.1.0 on the ISPD2005 dataset. The results in table below indicate that EGPlace outperforms Dreamplace 4.1.0 on 7 out of 8 benchmarks, achieving an average HPWL improvement of 25.1%.
>
> ## Table: Comparison to Dreamplace 4.1.0 on Macro Place over ISPD2005.
> | Method        | Adaptec1 | Adaptec2 | Adaptec3 | Adaptec4 | Bigblue1 | Bigblue2 | Bigblue3 | Bigblue4 |
> |---------------|----------|----------|----------|----------|----------|----------|----------|----------|
> | EGPlace | 5.75| 37.99 | 60.01 | 54.45| 2.23 | 10.55 | 49.98 |59.73 |
> | DreamPlace|10.24| 31.14 | 62.63 | 63.78| 6.07| 14.20 | 77.81 |92.51 |
>
> **Typo on MaskPlace**
>
> We apologize for the typo in Table 5. The HPWL achieved by MaskPlace is 96.91 × 10⁵ rather than 9.69 × 10⁵. We will correct this error in the revised version.

---

> > ### Comment · Reviewer_nQ5x · 2025-04-05
> >
> > Thank you for highlighting that Dreamplace 4.1 performs better in mixed-size placement scenarios. Since mixed-size placement is more critical than macro placement in our application, I suggest we update Table 6 with these latest results for a more accurate comparison.
> >
> > Regarding macro placement specifically, could you elaborate on the efficiency differences between Dreamplace and EGPlace in terms of runtime?

---

> > > ### Author Response · Authors · 2025-04-06
> > >
> > > Thanks for your valuable feedback! We plan to update Table 6 with the results of DreamPlace 4.1.0 in the revised version.
> > >
> > > Regarding macro placement on the eight ISPD2005 benchmark circuits, the average runtime of DreamPlace 4.1.0 and EGPlace is 19.8 seconds and 2925 seconds, respectively. This demonstrates that DreamPlace 4.1.0 achieves high computational efficiency compared to EGPlace. We make more explanation on the results:
> > >
> > > 1. DreamPlace, as an analytical-based method, generally offers significantly higher computational efficiency than other types of approaches.
> > >
> > > 2. Analytical-based methods have certain limitations in placement quality. For example, they face challenges in handling non-differentiable objectives such as congestion, which can further degrade the final placement quality. Moreover, due to relaxed overlap constraints, modules may heavily overlap during the global placement stage, requiring substantial displacement during the legalization step to resolve overlaps—this often leads to increased wirelength. As shown in the comparison table included in our previous rebuttal, EGPlace outperforms DreamPlace 4.1.0 in terms of placement quality, **achieving better results on 6 out of 8 data set**. While EGPlace takes approximately one hour to complete macro placement, we believe that this runtime is a **reasonable trade-off** for achieving significantly better results.
> > >
> > > 3. As demonstrated in Table 1 of the manuscript, EGPlace provides much higher efficiency compared to state-of-the-art reinforcement learning and stochastic-based methods, achieving a **2.8× speedup over EfficientPlace and a 7.8× speedup over WireMask-EA**.
> > >
> > > 4. Additionally, please note that DreamPlace is primarily implemented in **C**, while EGPlace is implemented in **Python**, which is an interpreted language and generally much slower in execution speed.
> > >
> > > For mixed-size placement, as mentioned in our previous rebuttal, EGPlace still has room for improvement. We expect that with further enhancements, the method could achieve even better placement results.
> > >
> > > **Regarding the PPA evaluation results:**
> > >
> > > We appreciate the comments and concerns raised by **Reviewer nQ5x** and **Reviewer pQfq** regarding the evaluation of PPA metrics. We would like to respond to their points below.
> > >
> > > We attempted to conduct macro placement on the ariane133 dataset supported by OpenROAD by selecting the 256 largest modules. Following the two-stage flow described in Appendix D5 of the manuscript, we performed mixed placement using DREAMPlace and then carried out PPA evaluation using the code provided in ChipBench.
> > > During the experiments, we encountered several challenges:
> > >
> > > 1.	The experimental flow was relatively complex, involving both EGPlace and DREAMPlace for placement, as well as conversions between LEF/DEF and Bookshelf formats. We had to carefully ensure data consistency throughout the process，and handling dataset format conversions was a new challenge that was not encountered in our previous experiments.
> > >
> > > 2.	The files generated from our early-stage filtering and conversion caused issues during DREAMPlace legalization. We spent several days identifying and resolving this problem.
> > >
> > > 3.	We evaluate PPA using ChipBench on a server with Intel Xeon Silver 4210R CPU (2.40GHz). The evaluation process is extremely time-consuming, sometimes taking over 24 hours. This resulted in long waiting times for results and made it difficult to quickly debug and adjust errors due to the slow feedback.
> > >
> > > Consequently, a considerable amount of time is needed to complete this experiment.
> > >
> > > The program is currently running and has reached the detailed routing stage. If it completes successfully and the results become available, we will present the subsequent experimental outcomes via link  https://anonymous.4open.science/r/EAPlace-31A4.
> > >
> > > Until now, the output log (last 10 lines) of the PPA evaluation program we are currently running is as follows:
> > >
> > >     [INFO DRT-0194] Start detail routing.
> > >     [INFO DRT-0195] Start 0th optimization iteration.
> > >     Completing 10% with 12097548 violations.
> > >     elapsed time = 01:31:38, memory = 34433.00 (MB).
> > >     Completing 20% with 21157172 violations.
> > >     elapsed time = 02:45:07, memory = 37354.73 (MB).
> > >     Completing 30% with 22915019 violations.
> > >     elapsed time = 03:20:10, memory = 37285.90 (MB).
> > >     Completing 40% with 35208776 violations.
> > >     elapsed time = 05:28:14, memory = 45461.79 (MB).

---

### Official Review · Reviewer_WoFK · 2025-03-13

**Overall Recommendation:** 3

**Summary:**

This article presents a new mutation operator for evolutionary algorithms designed for the problem of macro placement. The new operator, the Greedy Repositioning Guided Mutation, constructs a set of good placements for a module and then randomly selects one. Compared to a traditional mutation operator, it therefore encourages good placement of the module, while still allowing for search. This mutation operator is combined with a standard evolutionary algorithm and tested on a standard benchmark in macro placement.

**Claims And Evidence:**

The main claim is that the introduced mutation operator improves search for chip configurations. This is demonstrated clearly by the comparison between the proposed method EGPlace and a similar evolutionary method WireMask-EA, as well as other baseline methods.

**Essential References Not Discussed:**

Geng, Zijie, et al. "LaMPlace: Learning to Optimize Cross-Stage Metrics in Macro Placement." The Thirteenth International Conference on Learning Representations.

Ochoa, Gabriela, Katherine M. Malan, and Christian Blum. "Search trajectory networks: A tool for analysing and visualising the behaviour of metaheuristics." Applied Soft Computing 109 (2021): 107492.

**Experimental Designs Or Analyses:**

The experiments appear well designed, using the ISPD2005 dataset and the Ariane RISC-V CPU benchmark. The method is compared against 4 baseline methods; the EGPlace method consistently improves over other methods. There are a few points of improvement:
+ ICCAD2015 is a more recent benchmark than ISPD2005 - what is the motivation for ISPD2005?
+ The number of baseline methods included is limited. Wiremask-EA compares with SP-SA, NTUPlace3, RePlace, DREAM-
Place, and Graph placement, DeepPR, MaskPlace. It would be especially helpful to include baselines from the RL literature, as this represents a significant part of the literature.
+ The results presented in Table 1 could be improved through statistical analysis; best results are marked in bold, but it is not indicated if they are significant, nor for how many independent trials the comparison is performed.
+ Code is not included.

**Methods And Evaluation Criteria:**

The proposed method is a mutation operation for the problem of macro placement. In that context, it is appropriately evaluated, however in the "Experimental Designs Or Analyses" section, I provide some improvements to the experimental analysis which would strengthen the claims.

**Other Comments Or Suggestions:**

None.

**Other Strengths And Weaknesses:**

The article positions the contribution as a "novel evolutionary framework," however the main contribution is the mutation operator, as confirmed by the ablation study. At what point could other parts of the evolutionary algorithm be specified for this problem? It is surprising that a fitness-proportionate selection is used, for example; in genetic algorithms, tournament selection is now standard. Was elitism or truncation selection explored? Is recombination possible? A greater study of the evolutionary algorithm's details that engages with recent evolutionary literature would improve the article, especially if the contribution is intended to be the evolutionary framework in full and not only the mutation operator.

**Questions For Authors:**

None.

**Relation To Broader Scientific Literature:**

The article places itself well in the literature on the macro placement problem. It does not engage as fully with the evolutionary literature. For example, visualizing the different search trajectories taken by EGPlace compared to WireMask-EA would be a useful demonstration of the benefits of this new mutation operator.

**Theoretical Claims:**

There are no theoretical claims.

---

> ### Author Rebuttal · Authors · 2025-04-01
>
> Thanks for your valuable suggestions!
>
> **Motivation for choosing ISPD2005 and Additional Experiments on ICCAD2015**
>
> The major competitors, including MaskPlace, EfficientPlace, and WireMask-EA, conduct experiments on ISPD2005, so we use the same setting for a fair comparison. We also perform experiments on ICCAD2015, with results shown in Table (a) (see feedback to Reviewer pQfq), where the HPWL for placing the largest 1024 macros is reported and will be included in the revised version.
>
> **Response Regarding Baseline Methods**
>
> We omitted baselines SP-SA, NTUPlace3, RePlace, DREAMPlace, Graph Placement, and DeepPR, as they have been compared with our main competitors, MaskPlace and WireMask-EA, in their respective papers, where our competitors outperformed them. We plan to include results from these baselines in revised version for a more comprehensive comparison.
>
> **Statistical Analysis for Table 1**
>
> We conducted 5 independent trials, consistent with comparison methods, and report results as mean ± standard error. This will be included in the caption of Table 1 in revised version.
>
> A Wilcoxon rank-sum test was performed based on 5 trials. The symbols “+,” “–,” and “≈” indicate where the baseline method’s HPWL is significantly better, worse, or similar to EGPlace at a 0.05 significance level. The analysis is based on results from the released code. Due to time constraints, we ran WireMask-EA and EfficientPlace with 5 seeds on bigblue2 and bigblue4, so only partial results are reported. Additional results with 5 seeds will be included in revised version.
>
> The results show that EGPlace outperforms MaskPlace and Chipformer on all 8 benchmarks (confidence probability 0.0061). It also outperforms EfficientPlace on "bigblue2" and "bigblue4," and WireMask-EA on "bigblue4" (confidence probability 0.0061). EGPlace is statistically similar to WireMask-EA on "bigblue2" (confidence probability 0.2654).
>
> **Code is Not Included**
>
> We have uploaded our code into https://anonymous.4open.science/r/EAPlace-31A4.
>
> **The Article does not Engage as Fully with the Evolutionary Literature**
> We primarily focus on improving macro placement quality and efficiency, using an evolutionary framework for efficient iterations and broader search space exploration. However, we do not delve into the specifics of evolutionary algorithms.
>
> The key differences between WireMask-EA and EGPlace are: 1. WireMask-EA uses random mutation, while EGPlace employs guided reconstruction-based mutation for better sample efficiency; 2. EGPlace improves efficiency by selectively repositioning modules, avoiding the overhead of full layout reconstruction.
>
> We appreciate the recommended paper and believe the tool could help visualize the search trajectories of EGPlace and WireMask-EA, aiding comparison. We plan to conduct such visualizations and analyze their differences in future work.
>
> **Discussion of Related Paper**
>
> We appreciate the recommended paper and plan to include LaMPlace in related work section of our revised version. We also analyze the relationship between LaMPlace and EGPlace in our response to Reviewer pQfq. The trajectory visualization tool in the second paper is useful, and we plan to incorporate visualizations in future work.
>
> **Response Regarding the Evolutionary Framework**
>
> Our key ideas are as follows: 1.Mainstream RL methods suffer from high training costs and limited global context in decision-making. 2. Greedy adjustments to the entire layout can efficiently improve results. 3.An evolutionary search framework enables efficient iterations and broader exploration of the search space. Based on this, we adopt an evolutionary framework, focusing on improving layout quality and efficiency, rather than the specifics of evolutionary algorithms. We will remove the term Novel in revised version to avoid confusion.
>
> We have considered various strategies regarding selection and recombination in genetic algorithms. We think recombination is challenging, as merging different layouts often causes significant overlap. Post-processing methods like legalization and greedy reconstruction can reduce overlap but may compromise our efficiency advantage. For selection, we observe that high-quality layouts require sufficient refinement. Both EGPlace and WireMask-EA use small populations—WireMask-EA with 1, EGPlace with 5—for broader exploration. Given this setup, we did not implement a complex selection process. We use fitness-proportionate selection to determine which layout undergoes mutations, ensuring better layouts are adjusted with greater possibility while maintaining exploration. Our focus is on a simple yet effective approach that balances efficiency and layout quality.
>
> In future work, we plan to explore whether recombination can be efficiently implemented without excessive overlap while improving layout quality. If recombination is used, a larger population may be needed for sufficient exploration, requiring more effective selection strategies.

---

> > ### Comment · Reviewer_WoFK · 2025-04-03
> >
> > Thank you for the clarifications. Could the authors please clarify the choice of evolutionary algorithm? I'm glad to see this referred to as "fitness-proportionate selection" in your response, as opposed to "fitness-guided selection", as "fitness-proportionate" is a known term in the evolutionary literature. However, I must note that this is not how most evolutionary algorithms work. Consider the $(1+\lambda)$ EA:
> >
> >     Initialize x randomly
> >     while not terminate
> >         x_p = x
> >         for i in [1,λ]
> >             x_i = mutate(x_p)
> >             if f(x_i) < f(x)
> >                 x = x_i
> >     return x_p
> >
> > Each iteration has a population which is evaluated, selected, and then used for modification. This same loop applies to genetic algorithms, where using a random selection method like fitness-proportionate or tournament for parent selection and a second selection method like truncation is common. As noted in my first review, better engagement with the evolutionary literature would benefit this article. It isn't just to add references - the evolutionary algorithm used here is founded on some choices that have been studied in over 30 years of literature. I'll note the explanation of the selection scheme from the paper:
> >
> > "This fitness-guided selection is based on the intuition that layouts with higher fitness are more likely to result in high-quality layouts after adjustment. It ensures a more effective search while still allowing for random exploration."
> >
> > This intuition makes sense. But it has also been shown that rank-based selection, rather than fitness-based, is helpful to search as it is invariant to search space transformations. So a fitting alternative to fitness-proportionate selection would be tournament selection. However, for small population sizes like the ones used here, it isn't clear to me than any selection scheme besides truncation is necessary. Truncation selection is implicitly done in the proposed method during the exclude() method. So my understanding of the evolutionary algorithm used here is:
> >
> >     Initialize x1...xλ randomly
> >     sort(x1...xλ, f(x1)...f(xλ))
> >     x_best = f(x1)
> >     while not terminate
> >         x = fp_select(x1...xλ)
> >         x_p = mutate(x)
> >         if f(x_p) < f(xλ)
> >             xλ+1 = x_p
> >             sort(x1...xλ+1, f(x1)...f(xλ+1))
> >             delete(xλ+1)
> >         if f(x_p) < f(x_best)
> >             x_best = x_p
> >     return x_best
> >
> > Is that the case? So fitness proportionate selection is used for parent selection and there is one new individual created per generation? The deletion of the worst individual per generation is like truncation selection, but isn't done in the same way.
> >
> > Below are some examples that could help ground the evolutionary algorithm:
> >
> > Blickle, Tobias, and Lothar Thiele. "A comparison of selection schemes used in evolutionary algorithms." Evolutionary Computation 4.4 (1996): 361-394.
> >
> > Jansen, Thomas, Kenneth A. De Jong, and Ingo Wegener. "On the choice of the offspring population size in evolutionary algorithms." Evolutionary Computation 13.4 (2005): 413-440.
> >
> > Doerr, Benjamin, Carola Doerr, and Franziska Ebel. "From black-box complexity to designing new genetic algorithms." Theoretical Computer Science 567 (2015): 87-104.
> >
> > Hansen, Nikolaus, et al. "Impacts of invariance in search: When CMA-ES and PSO face ill-conditioned and non-separable problems." Applied Soft Computing 11.8 (2011): 5755-5769.
> >
> > Could the authors try to clarify their evolutionary algorithm in terms of existing algorithms? Furthermore, could justification be given for not using standard methods, like the 1+1, $1+\lambda$, or a genetic algorithm with tournament selection?
> >
> > I'll note that basing the code on popular open-source evolutionary libraries like pymoo or daep would avoid this confusion.

---

> > > ### Author Response · Authors · 2025-04-05
> > >
> > > Thank you for your valuable comments! We appreciate your accurate understanding and professional perspective. Your description in the second algorithm exactly captures the core idea of our approach. As you have pointed out, our method selects a layout from the population based on its fitness and applies a mutation operation to generate one offspring. The offspring is then added to the population, and the least fit layout is subsequently removed. This removal process resembles truncation selection.
> > >
> > > In our framework, a standard evolutionary algorithm could certainly be employed. However, we chose a tailored algorithmic design guided by empirical observations of experimental results to better balance placement quality and computational efficiency. Our proposed method can be viewed as an extension of the (1+1)-EA strategy employed by the baseline method WireMask-EA, distinguished by its use of a larger population size. Notably, when the population size is set to 1, our method becomes equivalent to (1+1)-EA. The decision to increase the population size is motivated by experimental evidence. As shown in Figure 6 in the manuscript, using a moderate population size (e.g., 3–5) results in better performance than a population size of 1. We believe this improvement is due to the broader search space enabled by a larger population, allowing more individuals the opportunity to evolve and contribute to the search process, ultimately leading to better placement outcomes.
> > >
> > > Since our approach extends the (1+1)-EA by enlarging the population size (i.e., having more than one individual in the population), it requires appropriate mechanisms for selecting individuals to generate offspring and for retaining promising individuals within the population. Unlike the (1+λ)-EA, which generates multiple offspring per iteration through multiple mutation operations, or standard genetic algorithms with tournament selection, which select multiple individuals to produce offspring, our method selects only one individual from the population in each iteration and applies a single mutation to generate one offspring. This design choice aims to reduce computational overhead. We find that generating one offspring per iteration tends to be sufficient for obtaining good results. Furthermore, since both the parent and the offspring can remain in the population, they continue to have opportunities to evolve in subsequent iterations. Producing multiple offspring or selecting multiple individuals at each iteration may not be essential and could increase computational cost without providing significant additional benefits. After generating the offspring, we remove the least-fit individual from the population using a simple truncation-like strategy. Given the relatively small population size in our setting, we believe this straightforward approach is sufficient and that more complex selection strategies may not offer substantial additional benefits.
> > >
> > > We believe that our proposed method is better suited to the current setting and achieves a good balance between efficiency and performance. (1+1)-EA and (1+λ)-EA are well-suited for scenarios with a population size of one. Tournament selection may be more effective in larger populations, where selecting multiple individuals per iteration can offer more benefits. Therefore, we chose to adopt the current approach in our design.

---

### Official Review · Reviewer_pQfq · 2025-03-25

**Overall Recommendation:** 3

**Summary:**

This paper presents EGPlace, an evolutionary optimization framework for macro placement. EGPlace addresses these issues with two key parts: 1) a greedy repositioning: guided mutation operator that targets critical layout regions and 2) an efficient mask computation algorithm. Experimental results on ISPD2005 and Ariane CPU benchmarks show that EGPlace reduces wirelength compared to WireMask-EA and EfficientPlace while achieving speedups. It also performs well in congestion control and mixed-size placement scenarios.

**Claims And Evidence:**

Yes, the empirical claims are well supported. However, there are some concerns about the evaluation criteria.

**Essential References Not Discussed:**

There are many recent papers on reinforcement learning for chip placement [1-3], which I believe should be at least discussed if not be compared.

[1-2] can be categorized into RL for adjustment-based methods. [3] applies a learnable mask and achieves sota performance.

[1] Reinforcement Learning Policy as Macro Regulator Rather than Macro Placer. NeurIPS, 2024.

[2] Mixed-Size Placement Prototyping Based on Reinforcement Learning with Semi-Concurrent Optimization. ASPDAC, 2025.

[3] LaMPlace: Learning to Optimize Cross-Stage Metrics in Macro Placement. ICLR, 2025.

**Experimental Designs Or Analyses:**

The experiments and analyses on HPWL make sense to me.

My main concerns are regarding the evaluation and the need for more benchmarks.

**Methods And Evaluation Criteria:**

The proposed approach demonstrates promising efficacy in optimizing HPWL.

However, there are several key considerations:

1. PPA Evaluation Gap. While HPWL serves as an important proxy metric, numerous studies in EDA and ML have highlighted its limited correlation with final PPA performance (e.g., timing, routed length, etc). I strongly recommend incorporating PPA evaluations using established open-source platforms such as OpenRoad [1] or the framework described in [2]. These tools enable final PPA evaluation, which would substantially imrpove this paper.

2. Expanded Benchmark. Testing on a broader range of more recent industrial chip designs (e.g., ICCAD 2015) would strengthen the generalizability claims, providing a more robust validation of the algorithm's adaptability.

References:
[1] OpenRoad Project. https://github.com/the-openroad-project

[2] Benchmarking End-to-End Performance of AI-Based Chip Placement Algorithms. arXiv, 2024.

**Other Comments Or Suggestions:**

1. WireMask-EA is proposed at NeurIPS 2023 rather than 2024.

**Other Strengths And Weaknesses:**

Strengths
1. The proposed mutation operator make sense to me, which can significantly improve exploration efficiency and HPWL results compared to random mutations in WireMask-EA.
2. The efficient mask computation is also important to rapidly evaluate potential module positions, reducing computational complexity from quadratic to linear.
3. The framework illustration figure is impressive and clear.

Weaknesses
1. See the evaluations mentioned above.

**Questions For Authors:**

N/A

**Relation To Broader Scientific Literature:**

N/A

**Theoretical Claims:**

N/A

---

> ### Author Rebuttal · Authors · 2025-04-01
>
> Thank you for your constructive review and valuable suggestions! Below, we provide a detailed response to your questions and comments. If any of our responses fail to sufficiently address your concerns, please inform us, and we will promptly follow up.
>
> **Experiments on the ICCAD 2015 Benchmark**
>
> In order to show the generalizability of EPlace, we further conduct tests on 5 circuits on the ICCAD 2015 dataset, selecting 1024 macros with the largest sizes for placement. The macro placement results, compared to EfficientPlace, are presented in Table (a).
>
> ## Table (a): HPWL (×10⁷) for Macro Placement on the ICCAD 2015 Benchmark
> | Method          | Superblue1 | Superblue3 | Superblue4 | Superblue5 | Superblue7 |
> |--------------- |------------|------------|------------|------------|------------|
> | EfficientPlace | 77.59      | 34.33      | 84.93      | 539.40     | 29.05      |
> | EGPlace       | 12.47      | 29.47      | 17.60      | 78.57      | 29.21      |
>
> The results show that EGPlace outperforms EfficientPlace on 4 out of 5 benchmarks. The significantly higher HPWL of EfficientPlace on Superblue1, 4, and 5 may result from the poor placement of the first few modules in nodes close to the root in MCTS, which ultimate affects the overall placement quality. We plan to include the full results in the revised version.
>
> We also attempted macro placement using DreamPlace 4.1.0 on the BookShelf file but encountered failures during the legalization stage. We will continue to investigate this issue and, if time permits, report the full results on the ICCAD 2015 benchmark, along with results for DreamPlace and other baseline methods.
>
> **Discussion on Recent Related Work**
>
> We will incorporate related work [1-3] into the related work section. Both [1] and [2] train RL policies that iteratively adjust module locations, as refining modules on the full layout enables capturing more comprehensive state information, which is beneficial for improving layout quality. We share the same observations with these methods. However, unlike these methods, we adjust the layout using efficient heuristic rules combined with a certain degree of randomness instead of training RL agents. Specifically, we select modules based on their scores and reposition them greedily, while utilizing an evolutionary algorithm to maintain a pool of good layouts. The search strategy we used is simple but effective in obtaining good quality layouts. Additionally, our method provides significant advantages in terms of efficiency.
>
> LaMPlace [3] makes significant advancements in incorporating cross-stage PPA metrics to guide layout generation. It trains a predictor to estimate these metrics through offline learning and generates learnable masks to assist placement. Our method, EGPlace, which leverages an evolutionary framework and adjusts module locations through a greedy reconstruction-based operator, is orthogonal to LaMPlace. The learnable masks related to PPA metrics generated by LaMPlace can be integrated into our approach to guide module adjustment, potentially achieving better performance in terms of PPA metrics.
>
> **Response on PPA Evaluation**
>
> We have attempted to use OpenRoad to evaluate the PPA metrics over the past few days. However, as noted in the ChipBench paper, OpenRoad is not compatible with the ISPD2005 and ICCAD2015 datasets due to the lack of essential information (e.g., necessary design kits). Therefore, We have to switch to "ariane133" benchmark which is supported by OpenRoad. The evaluation process is time-consuming. Due to time constraints, we plan to upload our results once the evaluation is completed.
>
> **WireMask-EA is proposed at NeurIPS 2023 rather than 2024**
>
> We sincerely apologize for the typo. We will correct this issue in the revised version.

---

### Decision · Program_Chairs · 2025-05-01

**Decision:**

Accept (poster)

**Comment:**

The paper addresses a known bottleneck in EDA with a demonstrably effective yet conceptually simple idea that is easy for practitioners to adopt. While the study seems to provide limited improvement and not as comprehensive on PPA as desired, its quality/efficiency trade‑off and open implementation make it a worthwhile contribution to the audience interested in optimization methods for real‑world design problems. We highly encourage the authors to incorporate reviewers' feedback. I will recommend a weak acceptance.